# MoSA: Mosaic Shared Adaptation of Large Language Models

**Xiequn Wang**[*1]    **Zhan Zhuang**[*1,2]    **Shengda Luo**[3]    **Yu Zhang**[†1]

[1] Southern University of Science and Technology    [2] City University of Hong Kong

[3] Chinese Medicine Guangdong Laboratory

`{wangxiequn,shengdaluo1991,yu.zhang.ust}@gmail.com`
`12250063@mail.sustech.edu.cn`

## Abstract

We introduce MoSA, a new parameter-efficient fine-tuning (PEFT) method that replaces low-rank factorization with randomized, fine-grained sharing of weight updates. Each adapted weight matrix is constructed by broadcasting a small set of learned scalars over a fixed tessellation, a pre-defined group assignment of weight entries of the weight matrix, producing expressive changes under the same parameter budget as low-rank adaptation (LoRA). MoSA requires no architectural changes and can be merged into the base model for zero-overhead inference. Across diverse language understanding and generation tasks, MoSA matches or surpasses strong PEFT baselines under strictly matched budgets. Analyses and ablations indicate that non-local parameter sharing acts as an effective regularizer, and that grouping design and budget allocation govern the expressivity–efficiency trade-off. These results position MoSA as a simple, scalable alternative to LoRA. Our code is available at https://github.com/XiequnWang/MoSA-ICLR26.

## 1 Introduction

Large Language Models (LLMs), such as BERT (Devlin et al., 2019), GPT-3 (Brown et al., 2020), and LLaMA (Touvron et al., 2023a), are built on Transformers (Vaswani et al., 2017) and pretrained on web-scale corpora. In practical use, these pretrained models are typically adapted to downstream tasks via task-specific fine-tuning, which has driven rapid progress across NLP and beyond. However, updating all parameters is often impractical: the computation and storage costs are substantial, optimizer states and checkpoints scale with the model size, and maintaining separate fully fine-tuned copies hinders multi-task and on-device deployment (Ding et al., 2023). This motivates parameter-efficient fine-tuning (PEFT), which seeks to retain the benefits of full adaptation while learning only a small set of additional parameters and keeping the base model frozen.

Among PEFT methods, low-rank adaptation (LoRA) (Hu et al., 2022) is widely adopted. It freezes pretrained weights and injects trainable low-rank matrices, reflecting the hypothesis that useful weight updates have low intrinsic ranks. This low-rank prior has become a de facto assumption in LoRA and its variants (Liu et al., 2024; Wang et al., 2024; Hayou et al., 2024; Meng et al., 2024; Wang et al., 2025b). However, this assumption imposes a strict structural bottleneck: it confines the weight update to a small subspace, potentially limiting the model's ability to learn complex, high-rank patterns required for difficult tasks. While recent approaches like HiRA (Huang et al., 2025) attempt to alleviate this limitation by using Hadamard products to achieve high-rank updates, they inherently rely on the priors of the original weights, effectively confining the update to a specific, weight-dependent subspace. This raises a natural question: can we achieve parameter efficiency without being bound by low-rank constraints or pre-existing weight structures?

To answer this question, we introduce Mosaic Shared Adaptation (MoSA). The intuition behind MoSA draws directly from the art of mosaics, where complex imagery is constructed from a limited

---

[*]Equal contribution.
[†]Corresponding author.

palette of tesserae. Similarly, MoSA partitions the entries of the weight matrix into a fixed tessellation of disjoint groups, and controls each group with a single learnable scalar. We then broadcast these scalars to their assigned positions to construct the final update. Crucially, we set the tessellation to be randomized and spatially agnostic. This non-local grouping disrupts the short-range correlations found in weight matrices, acting as a regularizer that mitigates co-adaptation (Hinton et al., 2012). This allows MoSA to construct expressive, full-rank updates using a parameter budget strictly comparable to LoRA.

Beyond the conceptual framework, we develop a first-order analysis showing that size-balanced grouping minimizes the expected deviation from the unconstrained update, thereby formalizing the optimality of our design. Furthermore, to realize MoSA efficiently, we implement a custom segmented-reduction backward kernel. This kernel aggregates per-group gradients in a single pass using stable permutation, eliminating the need for atomic operations and significantly accelerating the backward pass.

Comprehensive experiments on commonsense reasoning, open-domain dialogue, and mathematical reasoning benchmarks show that MoSA consistently outperforms strong PEFT baselines such as LoRA, DoRA (Liu et al., 2024), and HiRA (Huang et al., 2025) under matched parameter budgets. Notably, MoSA achieves competitive performance using a parameter budget smaller than a rank-1 LoRA equivalent, matching baselines that require tens of times more parameters.

## 2 RELATED WORKS

Adapting large pretrained models has produced a broad family of PEFT techniques. A useful organizing view is to ask whether a method *adds* small task–specific parameters while freezing the backbone, or *reparameterizes* the weight update itself. We follow this taxonomy and position our method accordingly. A qualitative comparison appears in Table 1. Related work is detailed below.

**Additive PEFT Methods.** A major branch of PEFT involves inserting small, trainable modules into the frozen LLM. Adapter-based methods (Houlsby et al., 2019) pioneered this by adding compact neural networks between Transformer layers. More recent approaches focus on the input and activation space. Prompt Tuning (Lester et al., 2021) prepends continuous, trainable "soft prompt" vectors to the input, while Prefix Tuning (Li and Liang, 2021) inserts trainable prefixes into the hidden states of each layer, steering the model's behavior without altering its core weights.

**Reparameterization via Low-Rank Updates.** An influential alternative is to reparameterize the weight update itself. LoRA (Hu et al., 2022) is the canonical example, modeling the update $\Delta W$ as two smaller, low-rank matrices ($\Delta W = BA$). This factorization dramatically reduces the number of trainable parameters. Its success has inspired numerous variants, including QLoRA (Dettmers et al., 2023) for memory efficiency, AdaLoRA (Zhang et al., 2023) for adaptive budget allocation and DoRA (Liu et al., 2024) for magnitude-direction decomposition.

**High-Rank Methods.** Recently, researchers have started exploring beyond the low-rank constraint (Jiang et al., 2024; Huang et al., 2025). MoRA (Jiang et al., 2024) employs a square matrix to maximize the rank of the update within a fixed parameter budget, challenging the necessity of low-rank decomposition. HiRA (Huang et al., 2025), for instance, uses Hadamard products to achieve high-rank updates efficiently, suggesting greater expressive power is beneficial. Our work aligns with this direction but proposes a fundamentally different mechanism.

**Hashing Methods.** The core idea of parameter sharing in MoSA is conceptually related to techniques developed for model compression. The "hashing trick" or HashedNets (Chen et al., 2015; Nooralinejad et al., 2023) use a hash function to group network weights, forcing all weights in the same hash bucket to share a single parameter value. This significantly reduces storage for the entire model. However, these methods were designed to compress the entire pre-trained weight matrix $W_0$. In contrast, MoSA applies this principle of randomized grouping specifically to the fine-tuning update $\Delta W$, repurposing it as a PEFT strategy rather than a static compression tool. This distinction is crucial, as MoSA maintains the integrity of the base model while enabling efficient adaptation.

Table 1: Qualitative comparison of PEFT methods (✓: yes, ✗: no). MoSA highlights: high-rank expressivity under the same budget, per-scalar budget granularity, lossless merge for zero-latency inference, no architectural changes, and non-local sharing as regularization.

| Criterion | MoSA (Ours) | LoRA | DoRA | MoRA | HiRA | Prompt/Prefix | Adapters |
|---|---|---|---|---|---|---|---|
| High-rank expressivity at fixed budget | ✓ | ✗ | ✗ | ✗ | ✓ | − | − |
| Arbitrary budget granularity | ✓ | ✗ | ✗ | ✗ | ✗ | ✗ | ✗ |
| No change in the architecture | ✓ | ✓ | ✓ | ✓ | ✓ | ✓ | ✗ |
| Non-local parameter-sharing regularization | ✓ | ✗ | ✗ | ✗ | ✗ | − | ✗ |

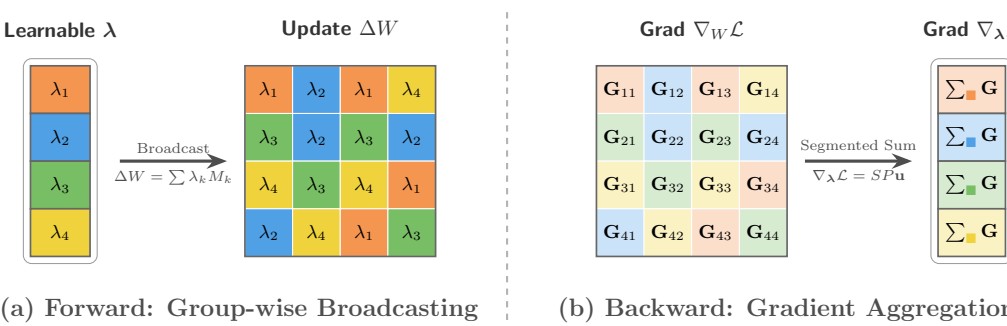

(a) Forward: Group-wise Broadcasting     (b) Backward: Gradient Aggregation

Figure 1: **MoSA Forward and Backward Mechanics. (a) Forward:** The learnable scalars $\lambda$ are broadcast to their corresponding color-coded tesserae to form the update matrix $\Delta W$. **(b) Backward:** The gradients $\nabla_W \mathcal{L}$ are aggregated via a segmented reduction. Values in identically colored cells are summed to produce the scalar gradient $\nabla_\lambda \mathcal{L}$, efficiently implemented as $\nabla_\lambda \mathcal{L} = SP\mathbf{u}$.

## 3 METHODOLOGY

In this section, we present **MoSA**. As illustrated in Figure 1, our method partitions the weight space into disjoint groups and assigns a single learnable scalar to each group. This design enables full-rank, element-wise modulation with a parameter budget comparable to or lower than that of low-rank adapters. We first detail the mathematical formulation, then provide a theoretical justification for our partitioning strategy, and finally describe the efficient gradient computation.

### 3.1 FORMULATION

Consider a pre-trained linear layer with weights $W_0 \in \mathbb{R}^{h \times d}$, where $d$ is the input dimension and $h$ is the output dimension. We aim to learn an additive update $\Delta W$.

**LoRA Parameterization.** LoRA assumes $\Delta W$ has a low intrinsic rank $r \ll \min(h, d)$. It parameterizes the update as the product of two low-rank matrices $B \in \mathbb{R}^{h \times r}$ and $A \in \mathbb{R}^{r \times d}$:

$$\Delta W_{\text{LoRA}} = BA. \tag{1}$$

This enforces a strict structural bottleneck, limiting the update to a low-dimensional subspace.

**MoSA Parameterization.** In contrast, MoSA constructs a full-rank update using a sparse set of shared scalars $\lambda \in \mathbb{R}^K$, where $K$ matches the parameter budget of LoRA baselines. We define a fixed partitioning of the weight indices $\mathcal{I} = \{(i, j) : 1 \le i \le h, 1 \le j \le d\}$ into $K$ disjoint sets (tesserae) $\mathcal{I}_1, \ldots, \mathcal{I}_K$. As shown in Figure 1(a), the update matrix is constructed via broadcasting:

$$\Delta W_{\text{MoSA}} = \sum_{k=1}^{K} \lambda_k M_k, \quad \text{where } (M_k)_{ij} = \begin{cases} 1 & \text{if } (i, j) \in \mathcal{I}_k, \\ 0 & \text{otherwise.} \end{cases} \tag{2}$$

Here, $M_k \in \{0, 1\}^{h \times d}$ serves as a binary mask for the $k$-th group. The forward pass for an input $x \in \mathbb{R}^d$ is given by $y = (W_0 + \Delta W)x$. Since the masks $\{M_k\}$ are mutually orthogonal and cover all indices, every weight element $W_{ij}$ is modulated by exactly one scalar $\lambda_k$. This allows MoSA to affect every weight individually while tying their values to a small set of learnable parameters.

**Gradient Derivation.** Let $\mathcal{L}$ denote the loss function. By the chain rule, the gradient with respect to the learnable scalar $\lambda_k$ is the Frobenius inner product between the weight gradient $\nabla_{\Delta W}\mathcal{L}$ and the mask $M_k$:

$$\frac{\partial \mathcal{L}}{\partial \lambda_k} = \langle \nabla_{\Delta W}\mathcal{L}, M_k \rangle_F = \sum_{(i,j)\in\mathcal{I}_k} \left( \frac{\partial \mathcal{L}}{\partial \Delta W} \right)_{ij}. \tag{3}$$

**Implementation via Matrix Permutation.** Although Eq. (3) provides the analytical form, we implement the backward pass as a fast linear projection in the vectorized space. Let $\mathbf{u} = \text{vec}(\nabla_{\Delta W}\mathcal{L}) \in \mathbb{R}^N$ be the flattened gradient vector, where $N = hd$. We define a fixed permutation matrix $P \in \{0,1\}^{N\times N}$ that reorders $\mathbf{u}$ such that indices belonging to the same group $\mathcal{I}_k$ become contiguous. As shown in Figure 1(b), the gradient aggregation is then expressed as:

$$\nabla_{\boldsymbol{\lambda}}\mathcal{L} = SP\mathbf{u}, \tag{4}$$

where $S \in \{0,1\}^{K\times N}$ is a *segmentation matrix* comprising contiguous blocks of ones. Specifically, $S$ maps the sorted segments of $P\mathbf{u}$ to their corresponding group scalars. In practice, the operation $SP$ is implemented as a custom fused segmented reduction kernel, which requires caching only the permutation indices, significantly reducing memory footprint compared to storing full adaptation matrices. Detailed explanations are shown in C.2.

**Structure Sharing.** To further minimize memory overhead, we enforce *structure sharing* across the network. Since modern architectures contain multiple layers with identical dimensions $h \times d$, we assign the same random partition to all weights of the same shape. This strategy allows us to store a single set of index mappings for each unique layer shape, decoupling the additional memory cost from the network depth.

### 3.2 THEORETICAL MOTIVATION: OPTIMALITY OF BALANCE

A critical design choice in MoSA is the distribution of group sizes $m_k = |\mathcal{I}_k|$. To understand the impact of group size, we examine the effective update dynamics. Let $\mathbf{G} = \nabla_W\mathcal{L} \in \mathbb{R}^{h\times d}$ denote the gradient of the loss with respect to the weights. MoSA updates scalar parameters $\lambda_k$ using the gradient $\nabla_{\lambda_k}\mathcal{L} = \langle \mathbf{G}, M_k \rangle_F$ and a learning rate $\eta$. Mapping this scalar update back to the weight space, the effective weight increment becomes:

$$\delta W_{\text{mosa}} = -\eta \sum_{k=1}^{K} \underbrace{\left( \sum_{(i,j)\in\mathcal{I}_k} \mathbf{G}_{ij} \right)}_{\text{Accumulated Gradient}} M_k = -\eta \sum_{k=1}^{K} m_k \bar{g}_k M_k, \tag{5}$$

where $\bar{g}_k$ is the mean gradient of the $k$-th group. Unlike standard projected gradient descent, which projects onto the mean direction $\bar{g}_k$, Eq. (5) reveals that the MoSA update is explicitly scaled by the group size $m_k$. This introduces an implicit, group-dependent learning rate scaling: larger groups receive aggressively larger updates. To ensure uniform optimization dynamics across all subspaces and minimize deviation from the global gradient direction, the scaling factor $m_k$ must be constant.

We formalize this intuition by quantifying the approximation error:

**Theorem 1** (Optimality of Balanced Partition). *Assume the entries of the gradient* $\mathbf{G}$ *are i.i.d. random variables with mean $\mu$ and variance $\sigma^2$. The expected squared error between the MoSA update and the unconstrained update is defined as:*

$$\mathcal{E}(m_1,\ldots,m_K) := \mathbb{E}\left[ \|-\eta\mathbf{G} - \delta W_{\text{mosa}}\|_F^2 \right]. \tag{6}$$

*For a fixed number of parameters $N$ and groups $K$, $\mathcal{E}$ is a Schur-convex function of the group sizes vector $\mathbf{m} = (m_1,\ldots,m_K)$. Consequently, $\mathcal{E}$ is minimized when the partition is balanced, i.e., $m_k \in \{\lfloor N/K \rfloor, \lceil N/K \rceil\}$.*

*Proof.* The squared error decomposes over the orthogonal basis formed by the group partitions. Since groups are disjoint, the total error is the sum of errors within each group:

$$\|\eta\mathbf{G} + \delta W_{\text{mosa}}\|_F^2 = \eta^2 \sum_{k=1}^{K} \|\mathbf{G}_{\mathcal{I}_k} - m_k\bar{g}_k M_k\|_F^2, \tag{7}$$

where $\mathbf{G}_{\mathcal{I}_k}$ denotes the restriction of $\mathbf{G}$ to indices in group $k$. Expanding the expected error term for a single group $k$, and noting that the MoSA update assigns the scalar value $\sum_{(i,j)\in\mathcal{I}_k}\mathbf{G}_{ij}=m_k\bar{g}_k$ to every element in that group:

$$\mathbb{E}\left[\sum_{(i,j)\in\mathcal{I}_k}(\mathbf{G}_{ij}-m_k\bar{g}_k)^2\right]=(m_k-1)\sigma^2+m_k(m_k-1)^2\mu^2+(m_k-1)^2\sigma^2. \quad (8)$$

Summing over all $k=1\ldots K$, the total expected error is:

$$\mathcal{E}(\mathbf{m})=\eta^2\left[(N-K)\sigma^2+\mu^2\sum_{k=1}^{K}m_k(m_k-1)^2+\sigma^2\sum_{k=1}^{K}(m_k-1)^2\right]. \quad (9)$$

The functions $\phi_1(x)=x(x-1)^2$ and $\phi_2(x)=(x-1)^2$ are convex for $x\geq 1$. Since the sum of convex functions applied to components of a vector is a Schur-convex function, $\mathcal{E}(\mathbf{m})$ is Schur-convex. By Karamata's inequality, a Schur-convex function is minimized when the components of its argument vector are as equal as possible. Thus, the error is minimized when $m_k\in\{\lfloor N/K\rfloor,\lceil N/K\rceil\}$ for all $k$. □

**Balanced Random Tessellation (BRT).** Theorem 1 implies that uniform group sizes maximize the fidelity of the update direction. Motivated by this, we implement *Balanced Random Tessellation* (BRT). We strictly enforce size constraints by randomly permuting the index set $\mathcal{I}$ and splitting it into $K$ contiguous blocks of equal size. By construction, BRT ensures that the gradient variance is spread evenly across all learnable scalars, maximizing training stability.

## 4 EXPERIMENTS

In this section, we evaluate MoSA across three distinct task families to demonstrate its effectiveness and generality: (i) multi-dataset commonsense reasoning, (ii) open-domain dialogue on ConvAI2 (Dinan et al., 2019), and (iii) mathematical reasoning under a distribution shift that involves training on MetaMathQA (Yu et al., 2024), evaluating on GSM8K (Cobbe et al., 2021). Throughout all experiments, we enforce strict *parameter-budget parity* with strong PEFT baselines. Our training and evaluation protocols are designed for transparency and reproducibility, closely following standard practices in recent literature (Huang et al., 2025) unless otherwise specified.

### 4.1 TASKS AND DATASETS

**Commonsense Reasoning.** We adopt a multi-task setting where models are trained once on the union of eight datasets and evaluated on each sub-task individually. The datasets include BoolQ (Clark et al., 2019), PIQA (Bisk et al., 2020), SIQA (Sap et al., 2019), HellaSwag (Zellers et al., 2019), WinoGrande (Sakaguchi et al., 2020), ARC-c (Clark et al., 2018), ARC-e (Clark et al., 2018), and OBQA (Mihaylov et al., 2018). The training sets are combined, comprising approximately 170k examples, with a small held-out split reserved for model selection. Evaluation is performed on the official test set of each respective dataset.

**Open-domain Dialogue.** We use the ConvAI2 dataset (Dinan et al., 2019), which consists of persona-grounded, multi-turn dialogues (17,878 for training, 1,000 for testing). The task is framed under the *self-persona* setting, where only the speaker's own persona is visible during generation. We report BLEU-4 (Papineni et al., 2002), BERTScore (P/R/F1) (Zhang et al., 2020), METEOR (Banerjee and Lavie, 2005), and ROUGE-L (Lin, 2004). We also present an Average score, calculated as the unweighted mean of these six metrics.

**Mathematical Reasoning.** For this task, we assess the model's out-of-distribution generalization by training on MetaMathQA (Yu et al., 2024) and evaluating on the GSM8K benchmark (Cobbe et al., 2021). We report exact match accuracy on the final numeric answer.

### 4.2 MODELS AND BASELINES

**Base Models.** We conduct experiments on two widely used open-weight models: Llama-2-7B (Touvron et al., 2023b) and Llama-3-8B (Grattafiori et al., 2024).

**Baselines.** We compare MoSA against a suite of strong PEFT methods, including Prompt Tuning (Lester et al., 2021), P-Tuning Liu et al. (2022), LoRA (Hu et al., 2022), DoRA (Liu et al., 2024), MoRA (Jiang et al., 2024), and the high-rank adaptation method, HiRA (Huang et al., 2025). All baselines are configured to adapt the same target modules as MoSA under matched training budgets.

## 4.3 Parameter Budget and Implementation

**Targeted Modules.** Unless stated otherwise, adapters are applied to the self-attention projections $\mathbf{W}_Q, \mathbf{W}_K, \mathbf{W}_V$ and the FFN projections $\mathbf{W}_{\text{up}}$ and $\mathbf{W}_{\text{down}}$ of each Transformer block. All other parameters in the base model are kept frozen.

**Parameter-Budget Parity.** To ensure a fair comparison, we enforce strict parameter parity between MoSA and LoRA at rank $r = 32$. For a target linear layer with dimensions $h \times d$, LoRA introduces $r(d + h)$ trainable parameters. Accordingly, we set the number of MoSA groups $K$ such that $K = r(d + h)$, guaranteeing an identical parameter budget for every target module. The column "Params (%)" in our tables reports this count as a percentage of the base model's total parameters.

**Optimization and Schedules.** We use the AdamW (Loshchilov and Hutter, 2019) optimizer with a learning rate of $1 \times 10^{-5}$ and 0.1 warm-up ratio. To ensure a fair comparison, we maintain consistency across all methods in maximum sequence length, tokenization, batch size, and mixed-precision settings. Models are trained for 3 epochs on the commonsense reasoning suite, 1 epoch on ConvAI2, and 2 epochs on the mathematical reasoning task. Model selection is performed on a held-out validation split. For all tasks, we employ deterministic decoding (temperature $= 0$).

## 4.4 Results: Commonsense Reasoning

As summarized in Table 2, MoSA demonstrates superior performance across the board on the multi-dataset commonsense reasoning benchmark. It consistently outperforms all strong PEFT baselines on both Llama-2-7B and Llama-3-8B, establishing itself as the most effective method for this task under strict parameter-budget parity.

On the more capable Llama-3-8B model, MoSA (with a budget equivalent to LoRA $r = 32$) achieves a remarkable average accuracy of 87.63%. This represents a substantial improvement of +0.91% over the strongest competing baseline, HiRA, which scored 86.72%. A closer look at the per-task results reveals MoSA's comprehensive dominance: it secures the top score on *every single one* of the eight datasets. The gains are particularly notable on challenging datasets like HellaSwag (+1.21% over the best baseline) and WinoGrande (+1.08%).

The same trend holds for Llama-2-7B. MoSA again leads the pack with an average accuracy of 83.83%, widening its lead over the second-best method, HiRA (81.42%), to a significant +2.41%. This confirms that MoSA's advantages are not specific to a single base model but are robust and transferable. On this model, it achieves the best performance on seven of the eight sub-tasks, further cementing its position as the state-of-the-art PEFT method for this reasoning suite.

## 4.5 Results: Open-domain Dialogue (ConvAI2)

In the persona-grounded, open-domain dialogue task on ConvAI2, MoSA's superiority is even more pronounced. As shown in Table 3, for Llama-3-8B, MoSA achieves an impressive result , surpassing the strongest baseline, HiRA (47.80%), by a large margin of +2.34%. This overall improvement is supported by consistent wins in all individual metrics, which measure different aspects of generation quality. For instance, its BLEU-4 score of 4.13% indicates significantly better n-gram overlap with reference responses, while its leading BERTScore (F1 of 86.83%) to superior semantic similarity and relevance. These results highlight MoSA's ability to produce responses that are more fluent and coherent (METEOR, ROUGE-L), while also being contextually and semantically more appropriate.

The results on Llama-2-7B further validate these findings. MoSA again claims the top spot on all seven metrics, culminating in an Average score of 49.92%. This represents an even larger improvement of +2.64% over the next-best method. The consistent and decisive lead across diverse automatic metrics underscores MoSA's enhanced ability to handle the nuances of multi-turn, persona-grounded conversations.

Table 2: Accuracy (%) on eight commonsense benchmarks using *Llama-2-7B* and *Llama-3-8B*. MoSA uses a parameter budget equivalent to LoRA rank $r$. Best is **bold**, second best is underlined.

| Method | Params (%) | BoolQ | PIQA | SIQA | ARC-c | ARC-e | OBQA | HellaSwag | WinoGrande | Average |
|---|---|---|---|---|---|---|---|---|---|---|
| *Llama-2-7B* | | | | | | | | | | |
| Prompt Tuning | 0.0012 | 55.93 | 12.35 | 30.50 | 6.06 | 8.63 | 9.40 | 6.91 | 40.57 | 21.29 |
| P-Tuning | 0.7428 | 58.75 | 36.02 | 0.20 | 0.17 | 1.98 | 0.80 | 0.01 | 0.00 | 12.24 |
| LoRA (r=32) | 0.8256 | 69.80 | 79.90 | 79.50 | 64.70 | 79.80 | 81.00 | 83.60 | 82.60 | 77.61 |
| DoRA (r=32) | 0.8256 | 71.80 | 83.70 | 76.00 | 68.20 | 83.70 | 82.40 | 89.10 | 82.60 | 79.69 |
| MoRA (r=32) | 0.8241 | 72.17 | 80.79 | 79.53 | 71.42 | 85.31 | 81.20 | 29.09 | 80.19 | 72.46 |
| HiRA (r=32) | 0.8256 | 71.22 | 83.35 | 79.53 | 73.81 | 86.74 | 84.60 | 88.12 | 83.98 | 81.42 |
| **MoSA (r=32 equiv.)** | 0.8256 | **73.96** | **86.03** | **81.48** | **76.62** | **88.34** | 83.93 | **94.35** | **85.93** | **83.83** |
| *Llama-3-8B* | | | | | | | | | | |
| Prompt Tuning | 0.0010 | 56.85 | 45.05 | 36.13 | 31.57 | 32.74 | 29.20 | 14.01 | 50.12 | 36.96 |
| P-Tuning | 0.6240 | 59.97 | 11.64 | 8.19 | 7.42 | 8.63 | 9.60 | 1.77 | 37.65 | 18.11 |
| LoRA (r=32) | 0.7002 | 70.80 | 85.20 | 79.90 | 71.20 | 84.20 | 79.00 | 91.70 | 84.30 | 80.79 |
| DoRA (r=32) | 0.7002 | 74.60 | 89.30 | 79.90 | 80.40 | 90.50 | 85.80 | 95.50 | 85.60 | 85.20 |
| MoRA (r=32) | 0.6997 | 74.28 | 87.43 | 80.71 | 79.61 | 91.16 | 85.60 | 43.53 | 86.74 | 78.63 |
| HiRA (r=32) | 0.7002 | 75.40 | 89.70 | 81.15 | 82.90 | 93.27 | 88.32 | 95.36 | 87.70 | 86.72 |
| **MoSA (r=32 equiv.)** | 0.7002 | **75.64** | **90.65** | **82.70** | **82.91** | **93.27** | **89.48** | **96.57** | **89.78** | **87.63** |

Table 3: ConvAI2 results with Llama-2-7B and Llama-3-8B backbones. BERT-F1/R/P are from BERTScore. Best is **bold**, second best is underlined.

| Method | Params (%) | BLEU-4 | BERT-F1 | BERT-R | BERT-P | METEOR | ROUGE-L | Average |
|---|---|---|---|---|---|---|---|---|
| *Llama-2-7B* | | | | | | | | |
| Prompt Tuning | 0.0012 | 0.04 | 72.44 | 77.38 | 68.23 | 0.80 | 0.80 | 36.62 |
| P-Tuning | 0.7428 | 0.60 | 83.29 | 83.33 | 83.28 | 15.11 | 12.36 | 46.33 |
| MoRA (r=32) | 0.8241 | 1.09 | 84.09 | 84.65 | 83.59 | 10.97 | 9.57 | 45.66 |
| LoRA (r=32) | 0.8256 | 1.82 | 84.41 | 84.71 | 84.16 | 11.38 | 10.55 | 46.17 |
| DoRA (r=32) | 0.8256 | 1.73 | 84.18 | 84.61 | 83.81 | 11.25 | 10.41 | 46.00 |
| HiRA (r=32) | 0.8256 | 2.70 | 84.86 | 84.98 | 84.77 | 13.56 | 12.80 | 47.28 |
| **MoSA (r=32 equiv.)** | 0.8256 | **3.93** | **86.53** | **86.28** | **86.80** | **17.57** | **18.44** | **49.92** |
| *Llama-3-8B* | | | | | | | | |
| Prompt Tuning | 0.0010 | 1.45 | 82.99 | 82.99 | 83.05 | 14.72 | 13.13 | 46.39 |
| P-Tuning | 0.6240 | 1.50 | 81.52 | 81.07 | 82.01 | 15.49 | 13.55 | 45.86 |
| MoRA (r=32) | 0.6997 | 1.60 | 82.24 | 84.06 | 84.43 | 12.37 | 11.19 | 46.31 |
| LoRA (r=32) | 0.7002 | 2.26 | 84.32 | 84.00 | 84.67 | 12.51 | 11.77 | 46.59 |
| DoRA (r=32) | 0.7002 | 2.29 | 84.32 | 84.06 | 84.62 | 12.63 | 11.78 | 46.62 |
| HiRA (r=32) | 0.7002 | 3.41 | 84.81 | 84.40 | 85.25 | 14.87 | 14.05 | 47.80 |
| **MoSA (r=32 equiv.)** | 0.7002 | **4.13** | **86.83** | **86.50** | **87.18** | **17.59** | **18.64** | **50.14** |

## 4.6 RESULTS: MATHEMATICAL REASONING

We evaluate the models on their out-of-distribution (OOD) generalization capabilities by training on MetaMathQA and testing on the unseen GSM8K benchmark. This challenging setup tests a model's ability to learn abstract reasoning principles rather than merely memorizing problem templates. As detailed in Table 4, MoSA demonstrates a profound and significant advantage in this area.

Using Llama-3-8B as the base model, MoSA achieves an exact match accuracy of 78.00% on GSM8K. This result is outperforming the strongest baseline, HiRA ($r = 32$), by a massive +7.19%. Such a large performance gap on an OOD task highlights MoSA's superior ability to capture and transfer the underlying logic of mathematical problem-solving. While other PEFT methods show respectable performance, MoSA's ability to generalize far more effectively sets it apart, indicating it learns more robust and portable reasoning structures.

Table 4: GSM8K accuracy (%) after training on MetaMathQA. Best is **bold**, second best is underlined.

| Method | Params (%) | GSM8K |
|---|---|---|
| *Llama-3-8B* | | |
| Prompt Tuning | 0.0010 | 15.62 |
| P-Tuning | 0.6240 | 2.65 |
| LoRA ($r = 32$) | 0.7002 | 65.89 |
| DoRA ($r = 32$) | 0.7002 | 66.12 |
| MoRA ($r = 32$) | 0.6997 | 67.98 |
| HiRA ($r = 32$) | 0.7002 | 70.81 |
| **MoSA ($r = 32$ equiv.)** | 0.7002 | **78.00** |

## 5 ANALYSIS

### 5.1 BACKWARD SPEED ANALYSIS

Since MoSA relies on aggregating gradients per group, the efficiency of the backward pass is paramount for scalability. Figure 2 illustrates the backward runtime (log–log scale) versus the number of groups $K$ for a weight matrix with dimensions $h = d = 4096$. We compare our custom *segmented reduction* kernel against PyTorch (Paszke et al., 2019) autograd under both *balanced* and *skewed* assignment strategies.

Across the entire range of $K$, our segmented reduction is consistently superior to autograd. Specifically, it achieves a $\approx 9{,}500\times$ speedup at $K = 1$ (0.25 ms vs. 2.41 s), a $125\times$ speedup at $K = 32$ (0.67 ms vs. 84.50 ms), and maintains an 8–9$\times$ advantage for $K$ between 4,096 and 16,384. Although skewness introduces a slight overhead (geometric-mean slowdown of $1.62\times$), the kernel remains highly efficient; even in the most extreme skew tested at $K = 65{,}536$, it is at least $4.8\times$ faster than autograd (1.46 ms vs. 7.00 ms). These results demonstrate that while balanced assignments (BRT) are optimal, they are not strictly required for performance: the segmented kernel is bandwidth-bound and robust to skew, supporting large $K$ without significant performance degradation. Additional experimental details are provided in Appendix B.

### 5.2 ABLATION STUDY ON COMPONENT

From the *Average* column in Table 5, FFN+QKV achieves the best overall accuracy (**87.63**), with FFN alone a close second (87.35) and marginally surpassing QKV (87.16). Here, FFN denotes the Transformer block's position-wise feed-forward network ($\mathbf{W}_{\text{up}}$ and $\mathbf{W}_{\text{down}}$), whereas QKV denotes the multi-head attention's query, key, and value projection sublayers/matrices ($\mathbf{W}_Q, \mathbf{W}_K, \mathbf{W}_V$). This indicates that most of the attainable gains under a fixed parameter budget come from adapting the feed-forward pathway, a finding consistent with evidence that FFN blocks function as key–value–like memories and host causally editable factual associations (Geva et al., 2021; Dai et al., 2022; Meng et al., 2022; 2023). Within attention-only variants, the ordering V > Q > K and QV > QK suggests that modifying the value stream—i.e., what content is written into the residual—is more impactful than adjusting query/key routing, aligning with mechanistic accounts where QK chiefly sets selection weights and V injects information (Elhage et al., 2021; Olsson et al., 2022). In practice, allocating most adaptation capacity to FNN and adding attention updates (especially V) for incremental gains is a robust default when budgets allow.

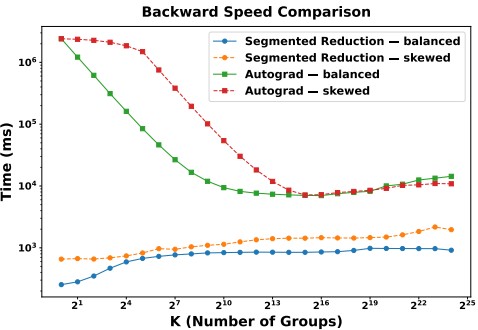

Figure 2: **Backward time vs. group count** $K$. Segmented reduction is faster for all $K$; skew induces a $1.62\times$ geometric-mean slowdown for segmented but does not alter the qualitative scaling.

### 5.3 IMPACT OF GROUPING STRATEGY

We evaluate whether *fine-grained, non-local* sharing provides stronger regularization than low-rank constraints by comparing our proposed BRT to three deterministic strategies defined below:

- **Balanced Random Tessellation (BRT)**: Set group sizes as balanced as possible, i.e., $m_k \in \{\lfloor N/K \rfloor, \lceil N/K \rceil\}$ such that $\sum_{k=1}^{K} m_k = N$. We generate a label sequence respecting these counts, uniformly permute the indices $\mathcal{I}$, and assign them to groups $\mathcal{I}_1, \ldots, \mathcal{I}_K$. This yields near-exact balance while breaking local spatial correlations.
- **Row-Stripe**: Flatten the weight matrix in row-major order and split the $N$ indices into $K$ contiguous segments. Indices in the $k$-th segment form the group $\mathcal{I}_k$. The result corresponds to $K$ horizontal stripes with sizes differing by at most one.
- **Col-Stripe**: Flatten in column-major order and split into $K$ contiguous segments. The result corresponds to $K$ vertical stripes with near-balanced sizes.

Table 5: Performance of the Llama-3 8B model with different Component. FFN denotes the position-wise feed-forward sublayer with projections $W_{\text{up}}$ and $W_{\text{down}}$

| Component | BoolQ | PIQA | SIQA | ARC-c | ARC-e | OBQA | HellaSwag | WinoG | Average |
|-----------|-------|------|------|-------|-------|------|-----------|-------|---------|
| K | 66.96 | 91.20 | 76.84 | 79.68 | 93.36 | 79.96 | 93.36 | 82.63 | 82.12 |
| Q | 70.45 | 92.13 | 79.08 | 80.53 | 94.43 | 84.72 | 94.43 | 85.06 | 84.33 |
| V | 74.85 | 92.72 | 82.24 | 81.63 | 95.64 | 87.50 | 95.64 | 86.48 | 86.29 |
| QK | 71.36 | 92.05 | 79.49 | 80.61 | 94.79 | 86.90 | 94.79 | 86.40 | 84.99 |
| QV | 74.85 | **93.48** | 82.60 | 82.82 | **96.17** | 87.90 | 96.17 | 87.50 | 86.81 |
| QKV | 75.46 | 92.97 | **82.91** | 82.74 | 96.06 | 89.29 | 96.06 | 87.81 | 87.16 |
| FFN | **76.13** | 90.98 | 81.94 | **83.5** | 93.18 | 86.71 | 96.46 | **89.86** | 87.35 |
| FFN+QKV | 75.64 | 90.65 | 82.70 | 82.91 | 93.27 | **89.48** | **96.57** | 89.78 | **87.63** |

- **Skewed**: Set non-increasing group quotas $m_1 \geq \cdots \geq m_K$ using a geometric schedule controlled by $\rho \in (0,1)$, where $m_k \propto \rho^{k-1}$. Uniformly permute the $N$ indices and assign the first $m_1$ to $\mathcal{I}_1$, the next $m_2$ to $\mathcal{I}_2$, and so forth. This preserves the intended skew while breaking local spatial correlations.

Table 6: Ablation of grouping strategies on Llama-3-8B under the same parameter budget.

| Strategy | BoolQ | PIQA | SIQA | ARC-c | ARC-e | OBQA | HellaSwag | WinoG | Average |
|----------|-------|------|------|-------|-------|------|-----------|-------|---------|
| BRT | 75.64 | 90.65 | 82.70 | 89.48 | 93.27 | 89.78 | 96.57 | 89.78 | **87.63** |
| Row-Stripe | 75.73 | 90.38 | 81.63 | 83.59 | 92.55 | 87.70 | 96.21 | 88.60 | 87.05 |
| Col-Stripe | 75.37 | 89.95 | 80.61 | 82.65 | 92.97 | 86.90 | 96.42 | 88.29 | 86.65 |
| Skewed | 61.22 | 81.52 | 70 | 71.68 | 85.19 | 69.64 | 86.49 | 69.18 | 74.36 |

From the Average column of Table 6, BRT performs best. Non-local randomization scatters sharing patterns and disrupts short-range correlations. Row-Stripe generally exceeds Col-Stripe because column-aligned sharing ties all outgoing connections of the same input feature, suppressing cross-row diversity and pushing outputs toward collinearity , whereas row-aligned sharing preserves more per-output specialization; more broadly, weight tying reduces degrees of freedom. Across layouts, skewed variants lag balanced ones: uneven group sizes concentrate capacity and increase collision variance, while balanced weight-sharing is empirically beneficial in hashing-style schemes. Implementation details appear in Appendix B.8.

## 5.4 Impact of Parameter Budget

We investigate how performance scales with the number of trainable tesserae $k$. A key advantage of MoSA is that $k$ can take any integer value, enabling fine-grained budget control unconstrained by matrix shapes, unlike LoRA. For reference, LoRA with rank $r=32$ on Llama-3-8B trains $0.7002\%$ of parameters (80.79 accuracy), which we use as a baseline.

As shown in Figure 3, accuracy rises steeply from extremely small budgets ($\sim 0.0014\%$). This corresponds to only about *one sixteenth* of the parameter count required by the smallest possible LoRA setting ($r=1$, $\sim 0.022\%$), even though LoRA itself cannot operate below rank 1. From there, performance continues to improve up to modest budgets ($\sim 0.175\%$, LoRA $r=4$ equivalent), after which the curve flattens. Notably, with an effective LoRA $r=1$ budget (0.022% parameters), MoSA already reaches **86.81**, surpassing the LoRA $r=32$ baseline while using only $\frac{1}{32}$ parameter numbers. Most gains are realized by the LoRA $r\approx4$ equivalent, with larger allocations bringing only marginal, task-specific improvements. Unlike LoRA, however, MoSA admits arbitrary $k$, allowing precise budget tuning between conventional ranks. Overall, these results suggest that small to moderate budgets (up to LoRA $r\leq4$ equivalents) already suffice for near-optimal accuracy, while larger budgets are only justified when additional task-specific gains are desired.

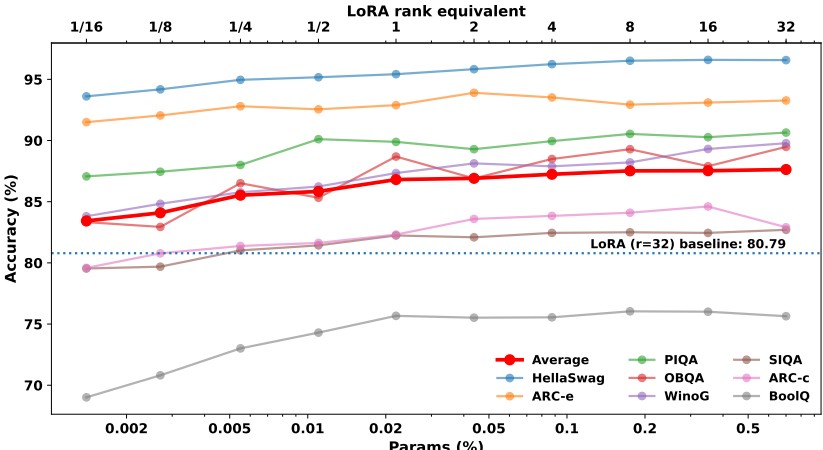

Figure 3: **Budget–performance trade-off on Llama-3-8B.** Per-task accuracy (thin *solid*) and Average (thick *dashed*) vs. trainable budget $k$ (log-scale, bottom axis). Top axis shows LoRA-rank equivalents; the vertical *dotted* line marks the LoRA $r=32$ baseline (80.79).

## 6 CONCLUSION

We introduced MoSA, a parameter-efficient fine-tuning method that replaces low-rank adapters with randomized, near-balanced tessellations of the weight matrix controlled by a small set of learned scalars. Theoretically, with uniform learning rates and balanced groups, the task-driven update is directionally equivalent to projecting the full gradient onto the MoSA subspace. System-wise, a segmented-reduction backward kernel aggregates per-group gradients in one pass, scales robustly with the number of groups, and delivers substantial speedups over autograd. Empirically, across commonsense reasoning, open-domain dialogue, and out-of-distribution math reasoning, MoSA consistently matches or surpasses strong baselines at the same parameter budget, indicating that simple non-local sharing is a competitive alternative to the low-rank assumption. Overall, MoSA offers a simple, controllable, and efficient route to parameter-efficient adaptation.

## ACKNOWLEDGMENTS

This work was supported by National Natural Science Foundation of China under Grant no. 62136005 and Shenzhen fundamental research program JCYJ20250604144724032. Shengda Luo was supported by the National Natural Science Foundation of China (Grant No. 82505358), the scientific research start-up funds of Chinese Medicine Guangdong Laboratory (Grant No. HQL2025SU011), and the TianYuan funds for Mathematics of the National Science Foundation of China (Grant No. 12326604).

## ETHICS STATEMENT

This work fine-tunes open-weight LLMs on public, widely-used benchmarks. We use the official splits and standard evaluation protocols; we do not collect new data, annotate human subjects, or process personally identifiable information. All datasets are used under their original licenses and terms. Public NLP datasets and the underlying pretrained models may contain harmful content or societal biases. Our method is a parameter-efficient adapter that modifies only a small set of scalars on top of frozen backbones, but it can still inherit and potentially amplify biases present in the data or base models. We therefore: (i) report task-standard metrics without promoting sensitive-attribute targeting; (ii) follow the base-model usage policies; and (iii) recommend downstream users apply appropriate safety filters, domain constraints, and bias audits when deploying adapted models in user-facing scenarios.

## REPRODUCIBILITY STATEMENT

We aim to make results reproducible. Detailed hyperparameter setting are shown in Appendix B. We have released our code at https://github.com/XiequnWang/MoSA-ICLR26.

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

# A    STATEMENT ON THE USE OF LLM

During the preparation of this manuscript, we utilized LLM solely for the purpose of grammar checking and language refinement. All core ideas, research content, data analysis, and conclusions are the original work of the authors. We take full responsibility for the final content of this paper.

# B    EXPERIMENTAL SETUP

## B.1    TASKS AND DATASETS

We evaluate on three task families to assess both in-domain performance and out-of-distribution generalization: (i) **Commonsense reasoning** on the union of *BoolQ*, *PIQA*, *SIQA*, *ARC-c/e*, *OBQA*, *HellaSwag*, and *WinoGrande* ($\approx$ 170k training examples in total); (ii) **Open-domain dialogue** on *ConvAI2* under the self-persona setting; (iii) **Mathematical reasoning (OOD)** by training on *Meta-MathQA* and evaluating on *GSM8K*. Unless stated otherwise, we follow standard dataset splits and report results on the official test sets with a small held-out validation split for model selection.

## B.2    BASE MODELS AND BASELINES

We consider open-weight LLMs Llama-2-7B and Llama-3-8B. Baselines include strong PEFT methods under matched conditions: Prompt Tuning, P-Tuning, LoRA, DoRA, MoRA, and HiRA. The results for these methods are directly copied from the HiRA paper with the same experimental conditions. All methods adapt the same target modules and use identical tokenization, maximum sequence length, batch size, and mixed-precision settings to ensure a fair comparison.

## B.3    ADAPTER TARGETS AND PARAMETER-BUDGET PARITY

Adapters are applied to the self-attention projections $\mathbf{W}_Q, \mathbf{W}_K, \mathbf{W}_V$ and feed-forward projections $\mathbf{W}_{\text{up}}, \mathbf{W}_{\text{down}}$ in each Transformer block, while other weights remain frozen. To enforce strict budget parity with a LoRA configuration of rank ($r = 32$), we choose the number of MoSA tesserae $K$ such that the total trainable scalars match $r(d_{\text{in}} + d_{\text{out}})$ for the same set of target matrices. We report *Params (%)* as the ratio of trainable parameters to the base model's total parameter count. Unless specified, group sizes are balanced (difference at most one), and the assignment is fixed throughout training.

## B.4    OPTIMIZATION AND TRAINING SCHEDULES

We use AdamW with learning rate $1 \times 10^{-5}$ and 0.1 warm-up ratio, cosine decay thereafter. Epochs per task: 3 for commonsense, 1 for ConvAI2, and 2 for MetaMathQA$\rightarrow$GSM8K. We train with a global batch size of 8, gradient accumulation 2, max sequence length of 512. Model selection is performed on the held-out validation split. Decoding uses deterministic settings (temperature $= 0$).

## B.5    EVALUATION PROTOCOL AND METRICS

For commonsense tasks, we report accuracy per dataset and the unweighted average. For dialogue, we report BLEU-4, BERTScore (P/R/F1), METEOR, and ROUGE-L, along with an unweighted average across metrics. For GSM8K, we report exact-match accuracy on the final numeric answer.

## B.6    HARDWARE AND REPRODUCIBILITY

Experiments are conducted on A100-80G PCIE. We fix random seeds, log all hyperparameters, and release scripts to reproduce results (data preprocessing, training, and evaluation).

## B.7    IMPLEMENTATION NOTES

MoSA parameterizes updates as groupwise-constant scalars over a fixed tessellation. Gradients are aggregated by a single segmented-reduction over a stable permutation that makes equal keys

contiguous. Balanced tessellation improves directional alignment of one-step updates and keeps segments near-uniform, aiding kernel utilization. Unless specified, the assignment is shared across layers of identical shape.

## B.8 DETAILS OF GROUPING STRATEGIES

In this section, we formalize the construction of the group-index map $\Gamma \in \{1, \ldots, K\}^{h \times d}$, which assigns each weight position $(i, j)$ to a group index $\Gamma_{ij} \in \{1, \ldots, K\}$. This map determines the group partition $\{\mathcal{I}_k\}_{k=1}^{K}$ and the corresponding mask matrices $\{M_k\}_{k=1}^{K}$ used in the main method, where

$$\mathcal{I}_k := \{(i, j) : \Gamma_{ij} = k\}.$$

Let $h$ and $d$ denote the output and input dimensions, respectively, and $N = hd$ be the total number of weights. We partition the index set into $K$ groups.

**Balanced Random Tessellation (BRT).** We aim to partition the $N$ weights into $K$ groups of nearly equal size. Let $r = N \bmod K$. We define the target group sizes as:

$$m_k = \begin{cases} \lceil N/K \rceil & \text{for } 1 \leq k \leq r, \\ \lfloor N/K \rfloor & \text{for } r < k \leq K, \end{cases} \tag{10}$$

which ensures $\sum_{k=1}^{K} m_k = N$ and $\max_k |m_k - N/K| \leq 1$.

We first generate a random permutation $\sigma$ of the linear indices $\{1, \ldots, N\}$ using a fixed seed. Let $\gamma \in \{1, \ldots, K\}^N$ denote the group-index assignment in linear form. The groups are formed by slicing this permuted sequence according to the sizes $\{m_k\}$: the $k$-th group $\mathcal{I}_k$ contains the indices $\{\sigma(t)\}$ for $t$ in the interval $(C_{k-1}, C_k]$, where $C_k = \sum_{j=1}^{k} m_j$ is the cumulative count. Finally, $\gamma$ is reshaped back to $(h, d)$ to form the map $\Gamma$.

---

**Algorithm 1** Construction of BRT Assignment

---

**Input:** Dimensions $h, d$, groups $K$, seed $S$
1: $N \leftarrow h \cdot d$
2: Compute sizes $\{m_k\}_{k=1}^{K}$ such that $m_k \approx N/K$             ▷ Balanced partition
3: $\sigma \leftarrow \text{RANDOMPERMUTATION}(\{1, \ldots, N\}; S)$
4: Initialize $\gamma \in \{1, \ldots, K\}^N$
5: $t \leftarrow 1$
6: **for** $k = 1$ **to** $K$ **do**
7:      **for** $j = 1$ **to** $m_k$ **do**
8:          Assign linear index $\sigma(t)$ to group $k$:     $\gamma_{\sigma(t)} \leftarrow k$
9:          $t \leftarrow t + 1$
10:      **end for**
11: **end for**
12: $\Gamma \leftarrow \text{reshape}(\gamma, h, d)$
13: **return** $\Gamma$

---

**Skewed Assignment.** To analyze the impact of load imbalance, we generate groups with non-uniform sizes following a geometric decay controlled by a ratio $\rho \in (0, 1)$. We define unnormalized weights $w_k = \rho^{k-1}$ for $k = 1, \ldots, K$. The target integer sizes $m_k$ are computed using the *Largest Remainder Method* to ensure they sum exactly to $N$:

1. Calculate ideal quotas $q_k = N \cdot (w_k / \sum_j w_j)$.

2. Assign initial integer sizes $m_k = \lfloor q_k \rfloor$.

3. Distribute the remaining count $R = N - \sum_k m_k$ to the $R$ groups with the largest fractional parts $q_k - \lfloor q_k \rfloor$.

The assignment is then performed similarly to BRT but using these skewed sizes on the random permutation. For our experiments, we set $\rho = 0.85$.

**Structured Striping Strategies.** Unlike the random assignment in BRT, stripe strategies preserve spatial locality. We define the cumulative boundaries $C_0 = 0$ and $C_k = \sum_{j=1}^{k} m_j$ based on balanced sizes $\{m_k\}$ as in BRT.

**Row-major stripes.** Weights are indexed in row-major order: $t(i, j) = (i - 1)d + j$. Indices falling in the interval $(C_{k-1}, C_k]$ are assigned to group $k$. Mathematically, $\Gamma_{ij} = k$ if and only if $C_{k-1} < (i-1)d + j \leq C_k$. This tends to group entire rows or contiguous row segments together.

**Column-major stripes.** Weights are indexed in column-major order: $t(i, j) = (j - 1)h + i$. Similarly, $\Gamma_{ij} = k$ if and only if $C_{k-1} < (j-1)h + i \leq C_k$. This tends to group entire columns or contiguous column segments.

---

**Algorithm 2** Construction of Structured Stripes (Row/Column)

---

**Input:** Dimensions $h, d$, groups $K$, mode $\in \{\text{ROW}, \text{COL}\}$
1: $N \leftarrow h \cdot d$
2: Compute balanced cumulative boundaries $\{C_k\}_{k=0}^{K}$
3: **for** $i = 1$ **to** $h$ **do**
4:     **for** $j = 1$ **to** $d$ **do**
5:         **if** mode is ROW **then**
6:             $t \leftarrow (i - 1)d + j$
7:         **else**                                        ▷ mode is COL
8:             $t \leftarrow (j - 1)h + i$
9:         **end if**
10:         Find $k$ such that $C_{k-1} < t \leq C_k$         ▷ Binary search or linear scan
11:         $\Gamma_{ij} \leftarrow k$
12:     **end for**
13: **end for**
14: **return** $\Gamma$

---

### B.9 Sensitivity to Random Seed

To evaluate the sensitivity of MoSA to the specific random tessellation generated by the seed, we conducted 5 independent runs using Llama-3-8B with the MoSA configuration. We used the same hyperparameters as the main experiments but varied the random seed used to generate the group assignments. As shown in Table 7, the performance is highly stable across different random initializations. The average accuracy across the five seeds is $87.45\%$ with a standard deviation of only $0.14\%$.

Table 7: Sensitivity analysis of MoSA (Llama-3-8B, $r = 32$ equivalent) across 5 different random seeds for tessellation assignment. The method shows high stability with minimal variance.

| Seed | BoolQ | PIQA | SIQA | ARC-c | ARC-e | OBQA | HellaS | WinoG | Average |
|---|---|---|---|---|---|---|---|---|---|
| 1 | 74.7 | 91.06 | 81.85 | 83.63 | 93.46 | 90.23 | 96.5 | 88.98 | 87.55 |
| 2 | 74.79 | 90.79 | 81.80 | 84.62 | 93.50 | 89.45 | 96.67 | 89 | 87.58 |
| 3 | 75.30 | 90.79 | 81.50 | 83.47 | 92.93 | 88.48 | 96.88 | 88.75 | 87.26 |
| 4 | 75 | 89.82 | 81.85 | 84.05 | 93.26 | 88.87 | 96.8 | 89.14 | 87.35 |
| 5 | 75.27 | 90.25 | 81.96 | 83.39 | 93.05 | 90.43 | 96.71 | 88.83 | 87.48 |
| **Mean** | 75.01 | 90.54 | 81.79 | 83.83 | 93.24 | 89.49 | 96.71 | 88.94 | **87.45** |
| **Std Dev** | 0.27 | 0.5 | 0.17 | 0.51 | 0.25 | 0.84 | 0.15 | 0.15 | **0.14** |

### B.10 Comparison to More PEFT Methods

We compare MoSA with PiSSA, HD-PiSAA, LoRA-GA and LoRA-Pro for two experiments. We utilized the Llama-2-7B (Touvron et al., 2023b) model as the backbone.

Regarding the datasets, we fine-tuned the model on a 100k subset of the MetaMathQA (Yu et al., 2024) dataset for the mathematical reasoning task and evaluated it on GSM8K (Cobbe et al., 2021).

For the code generation task, we fine-tuned on a 100k subset of the CodeFeedback (Zheng et al., 2024) dataset and reported the Pass@1 metric on HumanEval (Chen et al., 2021). The results for PiSSA (Meng et al., 2024), HD-PiSAA (Wang et al., 2025a), LoRA-GA (Wang et al., 2024), and LoRA-Pro (Wang et al., 2025b) were directly copied from their respective original papers.

Table 8: Fine-tuning results of Llama-2-7B model. **Bold** and underline indicate the highest and second-highest scores, respectively.

| Method | GSM8K | HumanEval |
|---|---|---|
| LoRA | 42.08 | 14.76 |
| PiSSA | 53.22 | 21.92 |
| HD-PISSA | 52.92 | 21.3 |
| LoRA-GA | 53.60 | 19.81 |
| LoRA-Pro | 57.57 | 22.97 |
| MoSA | **59.8** | **24.39** |

**Comparison with Multi-Task LoRA Methods.**    Although our primary experimental setting treats the aggregation of commonsense datasets as a single-task learning objective, the benchmark inherently consists of 8 distinct reasoning tasks. To validate MoSA's capability in this context, we compared it with HydraLoRA (Tian et al., 2024), a state-of-the-art method specifically designed for multi-task PEFT.

For the HydraLoRA configuration, we set the number of adapters $k = 8$ to align with the eight tasks and maintained the rank at $r = 32$. It is crucial to note that due to its multi-branch architecture, the actual trainable parameter count of HydraLoRA is approximately **4.5×** that of MoSA (or standard LoRA). The results are presented in Table 9. HydraLoRA achieves an average accuracy of **87.60%** across the eight datasets. Despite the significantly larger parameter budget, its performance is highly comparable to MoSA, which achieves an average of 87.66%. This demonstrates that MoSA effectively captures complex multi-task knowledge distributions through its sparse activation mechanism, without requiring the explicit routing structures or the massive parameter overhead typical of specialized multi-task methods.

Table 9: Performance comparison between MoSA and HydraLoRA ($k = 8, r = 32$) on commonsense reasoning tasks.

| Method | Params (%) | BoolQ | PIQA | SIQA | ARC-c | ARC-e | OBQA | HellaSwag | Wino | Avg. |
|---|---|---|---|---|---|---|---|---|---|---|
| MoSA | 0.70 | 75.64 | 90.65 | 82.70 | 82.91 | 93.27 | 89.48 | 96.57 | 89.78 | 87.63 |
| HydraLoRA | 3.60 | 75.03 | 91.11 | 82.46 | 84.21 | 93.71 | 88.28 | 96.90 | 89.06 | 87.60 |

### B.11    SENSITIVITY ANALYSIS OF LEARNING RATE

To evaluate the robustness of MoSA with respect to hyperparameter settings, we conducted a sensitivity analysis on the learning rate. We utilized the Llama-3-8B model with a rank of $r = 32$ and evaluated performance across eight standard commonsense reasoning benchmarks. We varied the learning rate within the range of $\{5 \times 10^{-6}, 1 \times 10^{-5}, 2 \times 10^{-5}, 5 \times 10^{-5}\}$, keeping all other hyperparameters constant.

The results are summarized in Table 10. We observe that MoSA exhibits stable performance across a broad range of learning rates. Specifically, the performance peaks at a learning rate of $1 \times 10^{-5}$, achieving the highest average accuracy of **87.63%** and leading in 5 out of 8 individual tasks. Furthermore, even when the learning rate deviates to $5 \times 10^{-6}$ or $2 \times 10^{-5}$, the fluctuation in average accuracy remains within 0.55%, demonstrating that our method is not overly sensitive to learning rate variations.

Table 10: Ablation study on learning rate sensitivity using Llama-3-8B ($r = 32$). **Bold** indicates the best result.

| Learning Rate | BoolQ | PIQA | SIQA | ARC-c | ARC-e | OBQA | HellaSwag | WinoGrande | Average |
|---|---|---|---|---|---|---|---|---|---|
| 5e-6 | 75.09 | 89.98 | 82.66 | **84.13** | **93.38** | 88.28 | 96.18 | 88.91 | 87.33 |
| 1e-5 | **75.64** | **90.65** | **82.70** | 82.91 | 93.27 | **89.48** | 96.57 | 89.78 | **87.63** |
| 2e-5 | 75.06 | 89.76 | 80.70 | 83.80 | 93.13 | 88.09 | **96.58** | **89.84** | 87.12 |
| 5e-5 | 74.52 | 89.28 | 80.85 | 82.65 | 91.69 | 87.89 | 95.52 | 87.27 | 86.21 |

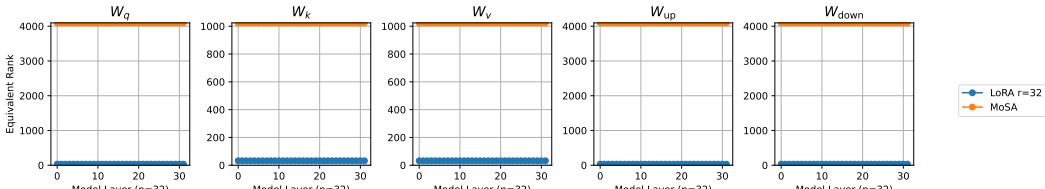

Figure 4: **Rank of Weight Updates.** We compare the effective rank of $\Delta W$ learned by LoRA and MoSA on LLaMA-3-8B. LoRA is strictly bounded by its low-rank bottleneck . In contrast, MoSA yields a full-rank update, utilizing significantly more dimensions in the weight space to capture complex features.

### B.12 RANK ANALYSIS

We calculate the rank update of MoSA and LoRA trained in LLaMA-3-8B on the commonsense dataset. We then compute the rank by SVD decomposition for singular values exceeding a threshold of $0.01$. As shown in Figure 4, LoRA is mathematically constrained to a low-rank subspace, limiting the number of active dimensions to its pre-defined rank $r$. Conversely, MoSA achieves a full-rank update.

## C SPEED AND GPU MEMORY USAGE

### C.1 SPEED

We benchmark the step time and throughput of MoSA against three strong parameter-efficient baselines: LoRA, DoRA, and HiRA. All methods are implemented in the same codebase and run on the same hardware with sequence length $512$, batch size $8$, and adapter rank $r = 32$. We use Adam as the optimizer and report the average wall-clock time per training step and tokens-per-second over $50$ iterations, after discarding warm-up steps.

Table 11 summarizes the results on Qwen3-4B-Base and Llama3-8B. On both models, LoRA attains the highest throughput, while our optimized MoSA implementation remains within the same order of magnitude as LoRA/DoRA/HiRA and is consistently much faster than the naive MoSA-Autograd variant, which uses the default Pytorch backward implementation. The latter serves as an ablation that highlights the benefit of our segmented-reduction implementation.

### C.2 MEMORY

For 3.1, although we formulate the gradient aggregation in Eq. 4 as a matrix-vector product $\nabla_\lambda \mathcal{L} = SPu$, explicitly storing the matrices $S$ and $P$ is computationally inefficient. Our implementation utilizes the BRT structure to represent these operators implicitly, reducing memory complexity from quadratic to linear.

**Implicit Permutation** ($P$). Instead of storing a permutation matrix $P \in \{0, 1\}^{N \times N}$, we store a single index vector $\boldsymbol{\pi} \in \mathbb{N}^N$ containing the sort order of the group assignments. The matrix-vector

Table 11: **Training speed comparison** sequence length 512, batch size 8, rank $r = 32$, Adam optimizer. We report average step time and throughput over 50 iterations.

| Model | Method | Step time ↓ (ms) | Tokens/s ↑ |
|-------|--------|------------------|------------|
| | LoRA | 360.07 | 11,375 |
| | HiRA | 386.12 | 10,608 |
| Qwen3-4B | DoRA | 440.63 | 9,296 |
| | MoSA(Ours) | 420.79 | 9,734 |
| | MoSA-Autograd | 1215.56 | 3,370 |
| | LoRA | 416.81 | 9,827 |
| | HiRA | 505.64 | 8,101 |
| Llama3-8B | DoRA | 532.81 | 7,688 |
| | MoSA(Ours) | 596.54 | 6,866 |
| | MoSA-Autograd | 2098.53 | 1,952 |

product $Pu$ is implemented as a direct memory gather operation, where the $i$-th element of the permuted vector is simply $u_{\pi_i}$. This reduces the storage requirement to $O(N)$.

**Implicit Segmentation** ($S$). The segmentation matrix $S \in \{0, 1\}^{K \times N}$ normally encodes which elements belong to which group. Under BRT, we enforce that all groups have a uniform size $m_k \approx N/K$. This regularity eliminates the need to store $S$. Since the permutation $P$ makes group elements contiguous, the aggregation $S(Pu)$ is mathematically equivalent to reshaping the permuted vector into a $K \times m_k$ matrix and computing the sum along the second dimension. This operation requires no additional metadata storage.

For the index matrix, we save $\Gamma$ shown in B.8. Therefore, the only additional cost is a small metadata buffer that stores the tessera index $\Gamma$ and sort order $\pi$ for each weight element to support the segmented-reduction kernel. This additional memory is a fixed constant once the partition is chosen. It depends only on the weight shapes and grouping, not on batch size, sequence length, optimizer, or rank $r$. On Qwen3-4B-Base and Llama3-8B, the buffer is about 470 MiB and 1088 MiB, respectively. This small overhead is largely due to the *structure sharing* strategy in Sec. 3.1, where layers with identical shapes reuse the same partition so that we store a single index buffer per unique shape instead of per layer.

# D ANALYSIS OF GENERALIZATION TO ADAPTIVE OPTIMIZERS

In this section, we extend our analysis from first-order SGD updates to adaptive optimization methods, specifically Adam, using the same notation as in Sec. 3.2. We show that balanced tessellations remain critical for maintaining uniform update magnitudes across different subspaces.

## D.1 GRADIENT STATISTICS UNDER AGGREGATION

Recall that MoSA updates a scalar parameter $\lambda_k$ that aggregates gradients from a group $\mathcal{I}_k$ of size $m_k = |\mathcal{I}_k|$. Let $\mathbf{G} = \nabla_W \mathcal{L} \in \mathbb{R}^{h \times d}$ denote the gradient, and write its entries as $\mathbf{G}_{ij}$. We assume the entries are i.i.d. with mean $\mu$ and variance $\sigma^2$:

$$\mathbb{E}[\mathbf{G}_{ij}] = \mu, \qquad \mathrm{Var}(\mathbf{G}_{ij}) = \sigma^2. \tag{11}$$

The gradient with respect to the scalar $\lambda_k$ is the Frobenius inner product between $\mathbf{G}$ and the mask $M_k$:

$$g_k := \nabla_{\lambda_k} \mathcal{L} = \langle \mathbf{G}, M_k \rangle_F = \sum_{(i,j) \in \mathcal{I}_k} \mathbf{G}_{ij}. \tag{12}$$

Thus the first and second moments of $g_k$ scale with the group size $m_k$:

$$\mathbb{E}[g_k] = m_k\mu, \tag{13}$$

$$\mathbb{E}\big[(g_k)^2\big] = \mathrm{Var}(g_k) + \big(\mathbb{E}[g_k]\big)^2 = m_k\sigma^2 + m_k^2\mu^2. \tag{14}$$

## D.2 EFFECTIVE UPDATE MAGNITUDE WITH ADAM

Adam updates parameters using the ratio of the first-moment estimate (denoted $\hat{m}_k$) to the square root of the second-moment estimate (denoted $\hat{v}_k$). For the scalar $\lambda_k$, at a steady state where the exponential moving averages approximate the corresponding expectations, the update $\Delta\lambda_k$ is approximately

$$\Delta\lambda_k \approx -\eta\frac{\mathbb{E}[g_k]}{\sqrt{\mathbb{E}\big[(g_k)^2\big]} + \epsilon} = -\eta\frac{m_k\mu}{\sqrt{m_k\sigma^2 + m_k^2\mu^2} + \epsilon}. \tag{15}$$

Mapping this scalar update back to the weight space, the effective increment for any weight in group $k$ satisfies

$$\delta W_{ij} = \Delta\lambda_k, \qquad \forall(i,j) \in \mathcal{I}_k. \tag{16}$$

We now analyze how the magnitude of this update scales with $m_k$ in two regimes.

**Regime 1: Variance-dominated.** In deep networks, gradient variance often dominates the mean ($\sigma^2 \gg \mu^2$) (Mori et al., 2022), especially in early training or for certain layers. When the variance term dominates the second moment, i.e., $\sigma^2 \gg m_k\mu^2$, we have

$$\delta W_{ij} \approx -\eta\frac{m_k\mu}{\sqrt{m_k\sigma^2}} = -\eta\frac{\mu}{\sigma}\sqrt{m_k}. \tag{17}$$

**Observation.** Even with Adam's normalization, the effective step size still scales as $\sqrt{m_k}$. If group sizes are unbalanced (e.g., one group is $4\times$ larger than another), the larger group will receive updates with approximately $2\times$ the magnitude.

**Regime 2: Signal-dominated.** If the mean gradient is strong, i.e., $\mu^2 \gg \sigma^2/m_k$, then the second-moment term is dominated by the squared mean:

$$\delta W_{ij} \approx -\eta\frac{m_k\mu}{\sqrt{m_k^2\mu^2}} = -\eta\,\mathrm{sgn}(\mu). \tag{18}$$

In this idealized regime, Adam successfully normalizes the update scale, and the dependence on $m_k$ vanishes. However, practical training typically oscillates between the variance-dominated and signal-dominated regimes.

As shown in Eq. (17), under the common variance-dominated condition the effective learning rate for group $k$ is proportional to $\sqrt{m_k}$. An imbalanced partition therefore implies that different parts of the weight matrix are optimized with effectively different learning rates. To ensure consistent optimization dynamics and stable moment estimation across all parameters, the group sizes $m_k$ must be kept approximately constant. This theoretically justifies the necessity of balanced tessellations even when using adaptive optimizers such as Adam.

