# OpenReview forum: "MoSA: Mosaic Shared Adaptation of Large Language Models"
_ICLR.cc/2026/Conference — ICLR 2026 Poster_

### Official Review · Reviewer_Vc9B · 2025-10-25

**Soundness:** 3
**Presentation:** 2
**Contribution:** 3
**Rating:** 6
**Confidence:** 4

**Summary:**

This paper introduces MOSA, a novel PEFT method that replaces LoRA's low-rank constraint with randomized, fine-grained parameter sharing. MOSA learns a single scalar update for pre-defined, size-balanced groups of weights (tessellations), enabling high-rank updates with no architectural changes or inference overhead. This balanced sharing acts as a strong regularizer, allowing MOSA to consistently outperform strong baselines like LoRA and HiRA in commonsense reasoning, dialogue, and out-of-distribution math tasks under matched budgets. The method is extremely parameter-efficient, notably surpassing a LoRA r=32 baseline while using only an r=1 equivalent budget.

**Strengths:**

- Superior Performance and Generalization: MOSA consistently outperforms strong PEFT baselines like LoRA, DoRA, and HiRA across diverse benchmarks, including commonsense reasoning and dialogue. Its advantage is particularly pronounced in out-of-distribution generalization, where it shows a superior ability to learn abstract reasoning principles.
- Extreme Parameter Efficiency: The method is highly efficient, achieving strong results at a fraction of the parameter cost. It can surpass the performance of much larger, well-established baselines while using only a very small budget, suggesting its sharing mechanism is a more effective use of parameters.
- Flexibility and Simplicity: It offers a simple alternative to LoRA by replacing the low-rank constraint with fine-grained parameter sharing. This design enables expressive, high-rank updates and provides highly granular budget control, as the number of trainable groups is not constrained by matrix shapes.
- Inference and Training Efficiency: MOSA requires no architectural changes and can be losslessly merged into the base model, resulting in zero-overhead inference. For training, it introduces a highly optimized segmented-reduction backward kernel that significantly accelerates gradient computation compared to standard autograd and scales robustly.

**Weaknesses:**

- `Confusing Terminology`: The term "Mosaic" fails to adequately reflect the method's core concept. I noted from the supplementary code that it was apparently previously named "Group-Share Adaptation." Clarification on the reason for this change is crucial for understanding the method and the paper's authenticity.
- `Limited Comparison`: The Commonsense Reasoning scenario employs a multi-task setup, yet the paper lacks a comparison against the wide range of existing MoE-based multi-task LoRA variants [1-3].
- `Unclear Presentation`: The paper's presentation is poor. For instance, Table 1 (referenced in Section 2) includes methods like DoRA that are never introduced, making the paper appear disjointed. The methodology section is inundated with formulas that feature arbitrary and unprofessional subscript notation. Furthermore, Figure 1 consumes excessive space while failing to convey sufficient methodological detail. The clarity of the paper's presentation is essential, and I will determine whether to update the final score based on the revised version.

[1] When MOE Meets LLMs: Parameter Efficient Fine-tuning for Multi-task Medical Applications

[2] HydraLoRA: An Asymmetric LoRA Architecture for Efficient Fine-Tuning

[3] CoLA: Collaborative Low-Rank Adaptation

**Questions:**

See Weaknesses.

---

> ### Author Response · Authors · 2025-11-24
> **Response to Reviewer Vc98**
>
> ### [W1] Confusing Terminology
> **Answer:**
>
> We apologize for the confusion caused by the mismatch between the name in the manuscript and the earlier internal name visible in the supplementary code.
>
> - **Naming history.** The method was initially developed under the internal working name *Group-Share Adaptation* (GSA), emphasizing that groups of weights share a scalar. During paper preparation, we renamed it to MoSA (Mosaic Shared Adaptation) to better reflect the tessellation metaphor and to avoid collision with other "group" terminology like group norm, group lasso, etc.
> - **Why "Mosaic."** The final name is directly tied to the core mechanism:
>   - Each weight matrix is partitioned into tesserae or tiles, exactly as in a mosaic art.
>   - A single scalar is shared within each tessera, and the overall update is assembled from many such tiles. Therefore we named it as "Mosaic Shared Adaptation."
>
> We made this connection explicit in the introduction (Line 041-042).
>
> ------
>
>
>
> ### [W2] Limited Comparison
> **Answer:**
>
> We agree that comparing against MoE-style multi-task LoRA variants is valuable, especially in the commonsense multi-task setting.
>
> - **HydraLoRA comparison.** In the revised version, we add a direct comparison to HydraLoRA, which is a strong and widely used multi-task LoRA variant:
>   - Backbone: Llama-3-8B.
>   - Scenario: the same 8-task commonsense reasoning benchmark used in our main results.
>   - Budget: HydraLoRA with $k=8$, $r=32$ and MoSA tuned to match our original PEFT budget.
>   - Results: HydraLoRA achieves comparable average accuracy to MoSA, but requires ≈4.5× more trainable parameters than MoSA because of having multiple LoRA heads per layer, whereas MoSA stays at ≈0.7% of backbone parameters.
> ------
>
>
>
> ### [W3] Unclear Presentation
> **Answer:**
>
> We appreciate the feedback about presentation. We have made several changes:
>
>
> #### Table 1 and missing introductions
>
> - We now introduce each baseline in Section 2.
>
> #### Notation and subscripts
>
> We agree that the notation in the methodology section was too dense. We have simplified notations by keeping most derivations in matrix form instead of switching aggressively to flattened vector indices, reducing the subscript clutter and avoiding non-standard symbols.
>
> #### Figure 1 redesign
>
> - We have redesigned Figure 1 to make it more informative and less space-wasting.
>
> We hope that these revisions substantially improve readability and address the reviewer’s concerns about clarity and professionalism of the presentation. We only change the presentation instead of method implementation.

---

> > ### Comment · Reviewer_Vc9B · 2025-11-26
> >
> > Thank you for your response. However, there are still issues with the wording. The core design of the method figure and its caption should fully reflect the paper’s motivation and immediately capture the reader’s attention. I’m sorry to say that this revised version still leaves me deeply confused. The theoretical analysis is encouraging, but theory should be built on clear, direct conclusions rather than adding cumbersome, unnecessary arguments just to make the method appear sophisticated. It is crucial to use simple and straightforward language to convey the motivation of the paper and the rationale behind the method’s design in order to convince the reviewers.

---

> > > ### Comment · Reviewer_Vc9B · 2025-11-26
> > >
> > > I genuinely believe that the quality of this paper exceeds the acceptance threshold of the ICLR conference. I apologize if my comments sometimes come across as sharp; as a reviewer, I am in fact acting mostly from the perspective of a reader. Subjectively, I feel that the paper does not provide sufficiently straightforward or detailed explanations of its motivation and the methods used to address that motivation. It might help to include some concrete real-world examples to aid understanding—something a reviewer may grasp, but an ordinary reader might find confusing.
> > >
> > > In the coming days, I recommend that the authors spend considerable time thinking carefully about offering more direct and detailed explanations (not necessarily in the main text—these could be placed in the appendix for clarity), rather than adding too much theory. This would give readers a sense of clarity and insight after reading the paper.

---

> > > > ### Author Response · Authors · 2025-11-26
> > > >
> > > > Thank you for your positive assessment and always constructive feedback. We are particularly grateful for your suggestion to adopt a reader's perspective, which is invaluable for improving our manuscript's quality.
> > > >
> > > > We take your advice to heart. In the coming days, we will focus entirely on making the presentation more straightforward and intuitive. Specifically, we plan to refine our motivation and add concrete examples to facilitate understanding, ensuring the paper offers clarity without being overburdened by theory. We strive to complete these revisions as soon as possible.
> > > >
> > > > Your suggestions have been immensely helpful to us.
> > > >
> > > > Best regards, The Authors

---

### Official Review · Reviewer_xciz · 2025-10-26

**Soundness:** 3
**Presentation:** 3
**Contribution:** 3
**Rating:** 4
**Confidence:** 4

**Summary:**

This paper introduces MoSA (Mosaic Shared Adaptation), a new parameter-efficient fine-tuning (PEFT) method for large language models (LLMs) that replaces LoRA’s low-rank decomposition with randomized, fine-grained parameter sharing. Instead of factorizing weight updates, MoSA partitions each weight matrix into a fixed number of groups (tesserae) and assigns a single learnable scalar per group, broadcast across all elements within it. This design enables high-rank expressivity under the same parameter budget as LoRA while maintaining full architectural compatibility and zero-overhead inference. The authors also propose an efficient segmented-reduction backward kernel that aggregates gradients per group without atomics, providing substantial training speedups. Empirically, MoSA is evaluated on commonsense reasoning (8 datasets), open-domain dialogue (ConvAI2), and mathematical reasoning (MetaMathQA→GSM8K) using Llama-2-7B and Llama-3-8B. Across all tasks, MoSA outperforms LoRA, DoRA, MoRA, and HiRA under matched parameter budgets (≈0.7–0.8% of model parameters).  A first-order theoretical analysis shows that balanced random tessellations minimize expected gradient discrepancy, formalizing why random grouping works as a regularizer.

**Strengths:**

1. Conceptual Novelty: MoSA breaks the low-rank assumption dominating PEFT (LoRA, DoRA) by proposing non-local, groupwise constant updates, a different structural prior.

2. Strong Empirical Evidence: The paper benchmarks across three families of tasks (reasoning, dialogue, OOD generalization) and two strong base models (Llama-2-7B, Llama-3-8B), all under strict budget parity. Results are consistent and robust.

3. Thorough Ablations: The study systematically probes grouping strategy, component selection, and budget scaling, reinforcing

**Weaknesses:**

1. Limited Theoretical Depth: The analysis in §3.1 (Theorem 1, in appendix) stops at first-order approximation. It does not address the *optimization dynamics* or convergence implications of groupwise sharing.

2. Potential Overclaim on High-Rank Expressivity: While empirically strong, the claim that MoSA can realize “high-rank updates” under small budgets is not theoretically substantiated (no explicit rank analysis of ∆W(λ)).

3. Lack of Comparison on Memory/Latency Trade-offs: The paper claims “zero-overhead inference,” but training-time memory and backward compute overhead are only partially reported (speed benchmarks but no GPU memory profiling).

4. Missing Larger-Scale or Real-World Fine-tuning Tasks: Only medium-scale reasoning and dialogue datasets are used; domain adaptation tasks (e.g., summarization, instruction tuning) are absent.

5. Presentation Minor Issues: The flattened notation and symbol reuse (e.g., Mh, Gh, g, π) are hard to follow for readers unfamiliar with vectorized representations.

**Questions:**

1. Expressivity Justification: Can the authors provide empirical evidence (e.g., singular value spectra) to demonstrate that MoSA indeed produces higher-rank ∆W updates than LoRA or DoRA at the same budget?

2. Gradient Projection Proof: Theorem 1 claims near-optimality of balanced tessellations; can the authors formalize this derivation in the main text and explain how it generalizes beyond first-order updates?

3. Ablation Granularity: How sensitive is MoSA’s performance to the random seed of tessellation assignment? Is grouping deterministic per layer or re-sampled per run?

4. Training Efficiency: What are the memory and time trade-offs versus LoRA for large-scale training (e.g., Llama-3-70B)? Is the backward kernel still efficient when scaling to multi-GPU distributed setups?

5. Interoperability: Could MoSA be combined with quantization (like QLoRA) or low-rank initialization (hybrid approaches)? Would grouping interact with quantized layers?

6. Generalization Analysis: Why does MoSA achieve such a large margin (+7% on GSM8K)? Any insights into whether the tessellation acts as regularization, or is this due to implicit gradient smoothing?

---

> ### Author Response · Authors · 2025-11-24
> **Response to Reviewer xciz**
>
> We appreciate the reviewer for the careful reading, positive assessment of the idea and experiments, and for the detailed suggestions. In the following response, we provide detailed answers to Weaknesses and Questions.
>
> ------
>
> ### [W1 & Q2] Limited Theoretical Depth & Gradient Projection Proof
> **Answer:**
>
> We agree that the theoretical analysis deserves greater prominence. In the revised version, we have significantly restructured the manuscript to highlight our contributions.
> - We have incorporated Theorem 1 and its proof into the main text.
> - We have extended our analysis beyond simple first-order SGD updates. In the new Appendix D, we have added results demonstrating that the main conclusions of Theorem 1 also hold for the Adam optimizer under variance-dominated settings.
>
> ------
>
> ### [W2, Q1 & Q6] Potential Overclaim on High-Rank Expressivity & Expressivity Justification
> **Answer:**
>
> To address this, we fine-tune Llama-3-8B on the commonsense dataset and examine the learned update $\Delta W$ for adapted matrices $W_Q, W_K, W_V, W_\text{up},\text{ and } W_\text{down}$ in the new Appendix B.12.
> - For each component in each layer, we compute the rank of $\Delta W$ via SVD, counting singular values above a threshold of 0.01.
>
> Findings:
> - For LoRA ($r=32$), the rank of $\Delta W$ is strictly bounded by $r$, and empirically stays near 32 for all layers, as expected from the low-rank parameterization.
> - For MoSA under the same parameter budget, the rank of $\Delta W$ is full across layers, meaning that MoSA effectively uses a large fraction of the available dimensions in weight space.
> - This high-rank expressivity, combined with the randomized tessellation acting as a regularizer to prevent reliance on structured priors, enables MoSA to capture complex reasoning patterns and achieve superior generalization (e.g., +7.19% on GSM8K).
>
> ------
>
> ### [W3 & Q4] Lack of Comparison on Memory/Latency Trade-offs & Training Efficiency
> **Answer:**
>
> For the **memory cost**, we have the following results.
>
> - Due to structure sharing (Sec. 3.3), layers with identical shapes reuse the same partition and thus the same index buffer, so the addtional memory cost depends only on the set of unique $(h,d)$, not on the network depth.
> - In our main training setting (seq=512, batch=8, $r=32$), the total additional memory cost(index matrices and sorting matrices) across the whole model is about 470 MiB for Qwen3-4B-Base and 1088 MiB for Llama-3-8B, and is independent of batch size, sequence length, optimizer, or rank $r$.
> - MoSA does not introduce extra activations beyond the base model and one scalar per tessera, so the activation and optimizer-state footprints are comparable to low-rank adapters.
>
> For the **latency**, we have added a direct end-to-end speed comparison with LoRA, DoRA, and HiRA (Appendix C, Table 11). All methods are implemented in the same codebase and run with sequence length 512, batch size 8, rank $r=32$, and Adam optimizer. We have the following reuslts.
>
> - On Qwen3-4B-Base, LoRA reaches 360 ms/step, whereas MoSA reaches 421 ms/step, comparable to DoRA (440 ms) and HiRA (386 ms).
> - On Llama-3-8B, LoRA is 417 ms/step, while MoSA is 597 ms/step, again in the same range as other high-rank baselines such as HiRA (506 ms/step) and DoRA (533 ms/step).
> - A naive MoSA-Autograd variant using vanilla PyTorch backward is 2–3× slower than our fused segmented-reduction kernel. This ablation study isolates the benefit of the specialized backward kernel.
> - Due to computational constraints, training full 70B models was not feasible.
> - Since MoSA relies exclusively on matrix operations, it incurs no additional asymptotic overhead in distributed environments.
>
>
> In summary, while LoRA is still the fastest, MoSA’s optimized implementation is within the same order of magnitude as LoRA/DoRA/HiRA in terms of the step time and throughput, and the additional memory is small.
>
>
>
> ------
> ### [W4] Missing Larger-Scale or Real-World Fine-tuning Tasks
> **Answer:**
>
> We agree that evaluating on diverse, real-world tasks is crucial for demonstrating the method's generality. To address this, we have added a Code Generation experiment in Appendix B.10, which serves as a challenging and common proxy for both domain adaptation (adapting from natural language to structured programming languages) and instruction tuning.
>
> Specifically, we utilized the a 100k subset of CodeFeedback dataset to fine-tune the Llama-2-7B model. The detailed information is shown in Appendix B.10 in the revised manuscript. As shown in Table 8 of Appendix B.10, MoSA achieves a Pass@1 of 24.39% on HumanEval, significantly outperforming LoRA (14.76%) and surpassing strong, recent baselines like LoRA-GA (19.81%), LoRA-Pro (22.97%) and PiSSA (21.92%), which demonstrates MoSA's effectiveness.

---

> ### Author Response · Authors · 2025-11-24
>
> ------
>
> ### [W5] Presentation Minor Issues
> **Answer:**
>
> We appreciate the feedback regarding notational clarity. In the revision, we have simplified the methodology section by keeping most derivations in matrix form rather than switching aggressively to flattened vector indices. This approach significantly reduces subscript clutter and minimizes the use of hard-to-follow symbols.
>
> We hope these revisions substantially improve readability and address the reviewer’s concerns. Please note that these revisions are purely presentational and do not alter the method as well as its implementation.
>
> ------
>
> ### [Q3] Tessellation randomness and sensitivity
> **Answer:**
>
> Regarding the granularity of ablations and sensitivity to the grouping, we have the following explanation.
>
> - For each unique layer shape, we sample a tessellation once using a fixed global seed and reuse it across all layers with that shape. The tessellation is kept fixed throughout training. It is not re-sampled per update or per epoch.
> - We conduct an ablation study where we only change the random seed used to generate the group assignments, and we observe only minor variation in aggregate metrics which are shown in Appendix B.9. This suggests that performance is robust to the particular random grouping.
>
> ------
>
> ### [Q5] Interoperability
> **Answer:**
>
> - Low-Rank Initialization: We have addressed this by explicitly comparing MoSA against strong initialization-based methods, including LoRA-GA, PiSSA, and HD-PiSSA, in the revised Appendix B.10. As shown in Table 8 of Appendix B.10, MoSA consistently outperforms these hybrid approaches.
>
>
> - Interaction with Quantization: Base model quantization (e.g., QLoRA) is orthogonal to MoSA. Since our grouping mechanism operates independently of base weight precision, MoSA is compatible with quantized layers and can theoretically serve as a drop-in replacement for LoRA. However, a dedicated "QMoSA" implementation is beyond the scope of this work.

---

### Official Review · Reviewer_xCoM · 2025-10-30

**Soundness:** 3
**Presentation:** 3
**Contribution:** 3
**Rating:** 6
**Confidence:** 2

**Summary:**

This paper introduces MoSA, a parameter-efficient fine-tuning method that replaces low-rank adaptation with fine-grained, randomized sharing of weight updates. Each weight matrix is adapted by broadcasting a small set of learned scalars over a fixed tessellation, enabling expressive updates without changing the model architecture and allowing zero-overhead inference. Experiments on diverse language understanding and generation tasks show that MoSA matches or exceeds strong PEFT baselines. Analyses indicate that non-local parameter sharing acts as a regularizer, and design choices in grouping and budget allocation control the expressivity–efficiency trade-off, positioning MoSA as a simple, scalable alternative to LoRA.

**Strengths:**

1. The proposed method is innovative and interesting.
2. The paper is well written and easy to follow.
3. The approach achieves strong empirical results, outperforming or equaling established PEFT baselines on diverse language modeling and generation benchmarks.

**Weaknesses:**

1. The work lacks theoretical analysis to explain why the proposed fine-grained, randomized weight-sharing mechanism is effective, leaving the underlying principles largely empirical.
2. The experiments are conducted only on the LLaMA series. Including evaluations on additional models, such as Qwen, would strengthen the empirical validation of the method.
3. MoSA is implemented only for linear layers and cannot be directly applied to convolutional layers, which may limit its generality across different model architectures.
4. How is the hyperparameter k controlled so that MoSA uses the same number of trainable parameters as LoRA with rank  r=32?

**Questions:**

Please see weaknesses.

---

> ### Author Response · Authors · 2025-11-24
> **Response to Reviewer xCoM**
>
> We thank the reviewer for the constructive feedback and the positive evaluation of our idea, writing, and empirical results. Below we address each weakness and question point-by-point, based on the new analyses and experiments in the revised manuscript.
>
>
> ------
>
> ### [W1] The work lacks theoretical analysis to explain why the proposed fine-grained, randomized weight-sharing mechanism is effective
> **Answer:**
>
> We apologize that the theoretical component was not sufficiently highlighted in the initial submission. The manuscript already contains a theoretical analysis of MoSA’s update dynamics and grouping strategy in Appendix, and we have clarified this more prominently in Section 3.2 of the revised version.
>
> - Section 3.2 analyzes MoSA as a constrained optimization in weight space. We derive the effective MoSA update:
>
>   $$  \delta W_{\text{MoSA}} = -\eta \sum_{k=1}^K m_k \bar{g}_k M_k,$$
>
>   where $m_k$ is group size and $\bar{g}_k$ is the mean gradient in group $k$.
>
>
> ------
>
> ### [W2] The experiments are conducted only on the LLaMA series
> **Answer:**
>
> We agree that testing multiple backbones improves the external validity of the method. We have conducted experiments on Qwen-family models.
>
> Here are the finetuning results on commonsense benchmarks for Qwen3-4B-Base under the same experimental setup as in the main manuscript. As shown in the following table, MoSA consistently outperforms LoRA on most datasets and improves the average accuracy from 86.08 to 87.78.
> |    Qwen3-4B-Base      | BoolQ     | PIQA      | SIQA      | ARC-c     | ARC-e | OBQA     | HellaSwag | WinoG     | Avg.      |
> | -------- | --------- | --------- | --------- | --------- | ----- | -------- | --------- | --------- | --------- |
> | LoRA     | 70.31     | 87.88     | 80.19     | **89.06** | 96.38 | 89.65    | 93.50     | 81.64     | **86.08** |
> | **MoSA** | **73.05** | **89.17** | **81.91** | 88.57     | 96.79 | **91.6** | **95.19** | **85.94** | **87.78** |
>
>
>
> ------
>
>
>
> ### [W3] MoSA is implemented only for linear layers and cannot be directly applied to convolutional layers
> **Answer:**
>
> Our primary focus in this work is Transformer-based language models, where the dominant trainable components are linear projections like attention and MLP. For that setting, restricting MoSA to linear layers already covers the majority of parameters and matches standard practice for LoRA-style methods.
>
> However, the MoSA formulation itself is not restricted to linear layers:
>
> - For a convolutional kernel $W \in \mathbb{R}^{C_\text{out} \times C_\text{in} \times K_h \times K_w}$, we can apply exactly the same tessellation mechanism after flattening the tensor to a vector. The whole procedure remains the same as linear layers.
> - Moreover, MoSA can also be adapted to parameters in normalization layers of CNNs.
>
> ------
>
>
>
> ### [W4] How is the hyperparameter k controlled so that MoSA uses the same number of trainable parameters as LoRA with rank r=32?
> **Answer:**
>
> We apologize for not stating this more explicitly in the original manuscript. The revised version now spells it out in Section 4.3 (Parameter-Budget Parity).
>
> For a target linear layer $W \in \mathbb{R}^{h \times d}$:
>
> - **LoRA with rank $r$** introduces two low-rank matrices $A \in \mathbb{R}^{h \times r}$ and $B \in \mathbb{R}^{r \times d}$. So the parameter count in LoRA is $r(h + d)$.
> - **MoSA** assigns one scalar $\lambda_k$ per group, so the number of trainable parameters is exactly $K$, the number of groups.
>
> To enforce strict per-layer budget parity with LoRA at rank $r=32$, we set $K$ as
> $$K = r(h + d).$$
>
> This ensures exact equality for each adapted layer. We apply MoSA and LoRA to the same set of target modules (self-attention $W_Q, W_K, W_V$ and FFN $W_\text{up}, W_\text{down}$), and so the total trainable parameter fraction reported in the tables is directly comparable across methods.

---

### Official Review · Reviewer_JoHz · 2025-11-01

**Soundness:** 2
**Presentation:** 2
**Contribution:** 2
**Rating:** 4
**Confidence:** 3

**Summary:**

This paper introduces MoSA, a PEFT method that constructs an adapted weight matrix by broadcasting a small set of learned scalars over a pre-defined group assignment of weight entries. MoSA features high-rank expressivity, arbitrary granularity, and non-local parameter-sharing regularization. MoSA is also coupled with efficiency techniques like Segmented Reduction and pre-caching to speed up training.              Experiments demonstrate MoSA’s performance advantage over LoRA, DoRA, and high-rank methods like MoRA and HiRA under the same updated parameter budget.

**Strengths:**

1. This paper proposes a novel PEFT method, first grouping original weight entries, then fine-tuning them with scaling parameters.                   This design ensures a high-rank update during fine-tuning.
2. MoSA’s realization is theory-grounded. The paper offers a theoretical view supporting its Balanced Random Tessellation design.
3. Performance improvement over baselines is significant.

**Weaknesses:**

1. Additional memory cost and latency. Since MoSA needs to precompute and cache (g, π, m, o) for each unique layer shape, it means MoSA must store more than two extra full-dimensional matrices, which are not present in methods like LoRA. Additionally, although the gather operation is optimized, it is still a slower operation than LoRA’s matrix multiply. This paper doesn’t include a direct speed comparison with LoRA.
2. Lacks stronger baselines. Stronger methods (e.g., LoRA-GA, LoRA-pro, PiSSA, HD-PiSSA…) are not compared. DoRA, MiRA, and HiRA are not strong baselines.
3. No updated rank analysis, which cannot support the paper’s high-rank expressivity claim.

**Questions:**

1. Could the authors provide the speed and memory comparison with LoRA?]
2. See other weaknesses above.
3. Is MoSA more sensitive to the learning rate or not?

---

> ### Author Response · Authors · 2025-11-24
> **Response to Reviewer JoHz**
>
> We thank the reviewer for the careful reading and constructive feedback. Below we address the raised concerns point by point, based on new experiments and analyses added in the revised version.
>
> ------
>
>
>
> ### [W1 & Q1] Additional memory cost and latency
>
> **Answer:**
>
> For the **memory cost**, we have the following results.
>
> - Due to structure sharing (Sec. 3.3), layers with identical shapes reuse the same partition and thus the same index buffer, so the additional memory (index matrices and sorting matrices) cost depends only on the set of unique $(h,d)$, not on the network depth.
> - In our main training setting (seq=512, batch=8, $r=32$), the total additional memory across the whole model is about 470 MiB for Qwen3-4B-Base and 1088 MiB for Llama-3-8B, and is independent of batch size, sequence length, optimizer, or rank $r$.
> - MoSA does not introduce extra activations beyond the base model and one scalar per tessera, so the activation and optimizer-state footprints are comparable to low-rank adapters.
>
> For the latency, we have added a direct end-to-end speed comparison with LoRA, DoRA, and HiRA (Appendix C, Table 11). All methods are implemented in the same codebase and run with sequence length 512, batch size 8, rank $r=32$, and Adam optimizer. We have the following results.
> - On Qwen3-4B-Base, LoRA reaches 360 ms/step, whereas MoSA reaches 421 ms/step, comparable to DoRA (440 ms) and HiRA (386 ms).
> - On Llama-3-8B, LoRA is 417 ms/step, while MoSA is 597 ms/step, again in the same range as other high-rank baselines such as HiRA (506 ms/step) and DoRA (533 ms/step).
> - A naive MoSA-Autograd variant using vanilla PyTorch backward is 2–3× slower than our fused segmented-reduction kernel. This ablation study isolates the benefit of the specialized backward kernel.
>
> In summary, while LoRA is still the fastest, MoSA’s optimized implementation is within the same order of magnitude as LoRA/DoRA/HiRA in terms of the step time and throughput, and the additional additional memory is small.
>
> ------
>
>
>
> ### [W2] Stronger baselines: LoRA-GA, LoRA-Pro, PiSSA, HD-PiSSA
> **Answer:**
>
> We agree that comparing against more recent PEFT methods is valuable. In the revision, we added new experiments with stronger baselines (Appendix B.10) with the following configurations.
> - Backbone: Llama-2-7B.
> - Tasks:
>   - GSM8K after fine-tuning on a 100k subset of MetaMathQA.
>   - HumanEval Pass@1 after fine-tuning on a 100k subset of CodeFeedback.
>
> The results in Table 8 are summarized as follows.
>
> - **GSM8K (↑):** LoRA 42.08, PiSSA 53.22, HD-PiSSA 52.92, LoRA-GA 53.60, LoRA-Pro 57.57, MoSA 59.80.
> - **HumanEval Pass@1 (↑):** LoRA 14.76, PiSSA 21.92, HD-PiSSA 21.30, LoRA-GA 19.81, LoRA-Pro 22.97, MoSA 24.39.
>
> Thus, MoSA outperforms all four methods on both mathematical reasoning and code generation in this setting. Detailed information is shown in Appendix B.10 of the revised manuscript.
>
> ------
>
>
>
> ### [W3] Rank analysis
> **Answer:**
>
> According to your suggestion, we added new experiments to analyze the rank. The experimental setup is described as follows.
> - We fine-tune Llama-3-8B on the commonsense dataset and examine the learned update $\Delta W$ for all adapted matrices ($W_Q, W_K, W_V, W_\text{up}, W_\text{down}$) in the new Appendix B.12.
> - For each layer and each module, we compute the rank of each $\Delta W$ via SVD, counting singular values above a threshold of 0.01.
>
> For the experiments, we have the following results.
>
> - For LoRA ($r=32$), the rank of $\Delta W$ is strictly bounded by $r$, and empirically stays near 32 for all layers, as expected from the low-rank parameterization.
> - For MoSA under the same parameter budget, the rank of $\Delta W$ is full across layers, meaning that MoSA effectively uses a large fraction of the available dimensions in the weight space.
>
> ------
>
> ### [Q3] Learning rate sensitivity
> **Answer:**
>
> We have added a dedicated learning-rate sensitivity ablation (Appendix B.11, Table 10).
>
> - Backbone: Llama-3-8B, MoSA with $r=32$ equivalent budget on the 8-task commonsense benchmark.
> - Learning rates tested: $5\times10^{-6}$, $1\times10^{-5}$, $2\times10^{-5}$, $5\times10^{-5}$, with all other hyperparameters fixed.
>
> **Results (average accuracy over 8 datasets):**
>
> - 5e-6: 87.33
> - 1e-5: 87.63 (the best)
> - 2e-5: 87.12
> - 5e-5: 86.21
>
> When varying the learning rate within a reasonable range, the change of the average accuracy stays within 0.55 percentage, indicating that MoSA is not so sensitive to this hyperparameter.

---

> > ### Comment · Reviewer_JoHz · 2025-11-26
> >
> > Thanks to the authors for the response. I appreciate the comprehensive experiments, which mostly solve my concerns and convince me that MoSA introduces a relatively acceptable extra cost compared to its superior performance. I have updated my scores accordingly to support this paper.

---

### Author Response · Authors · 2025-11-24
**General Response to All Reviewers**

Dear Reviewers and ACs:

We appreciate the diligent efforts and insightful comments from all reviewers and ACs. We have carefully considered your feedback to improve our manuscript. Please feel free to let us know if you have any further questions or require additional details.

In response to your valuable suggestions, we have revised the manuscript to strengthen the empirical evaluations, theoretical analysis, and clarity of presentation. We have included the main updates in the revised version:
- Table 8 (Appendix B.10): We have added experiments with stronger baselines (LoRA-GA, LoRA-Pro, PiSSA, HD-PiSSA) on mathematical reasoning and code generation tasks to demonstrate MoSA's robustness (suggested by Reviewers JoHz and xciz).
- Table 9 (Appendix B.10): We have added a comparison against MoE-based multi-task LoRA variants on the commonsense reasoning dataset (suggested by Reviewer Vc9B).
- Table 10 (Appendix B.11): We have conducted a sensitivity analysis on the learning rate to demonstrate that MoSA exhibits stable performance across a broad range of hyperparameter settings (suggested by Reviewer JoHz).
- Table 11 (Appendix C): We conducted a direct end-to-end speed and memory comparison with LoRA, DoRA, and HiRA to clarify training efficiency trade-offs (suggested by Reviewers JoHz and xciz).
- Figure 4 (Appendix B.12): We added a Singular Value Decomposition (SVD) analysis comparing the singular value spectra of weight updates learned by LoRA versus MoSA to substantiate high-rank expressivity claims (suggested by Reviewers JoHz and xciz).
- Appendix D: We extended our theoretical analysis to demonstrate that Balanced Random Tessellation (BRT) maintains optimality when using the Adam optimizer, going beyond the original SGD analysis (suggested by Reviewer xciz).
- Presentation Updates: We have redesigned Figure 1 and simplified the mathematical notation to improve readability and professionalism (suggested by Reviewers xciz and Vc9B).

We believe these revisions comprehensively address the reviewers' concerns.

Best regards,

The Authors

---

### Meta-Review · Area_Chair_8LgZ · 2026-01-04

**Summary:**

This paper presents MOSA, a parameter-efficient fine-tuning (PEFT) method that replaces LoRA's low-rank constraint with randomized, fine-grained weight sharing. By grouping weights into balanced "tessellations" and learning a single scalar update per group, MOSA achieves high-rank updates without architectural changes or inference cost. This design acts as a powerful regularizer, enabling MOSA to outperform strong baselines like LoRA and HiRA across reasoning, dialogue, and math tasks under matched parameter budgets.

The approach is both innovative and well-motivated. The authors are encouraged to incorporate the relevant reviewer feedback and associated discussion into the final manuscript to further strengthen the presentation.

**Reviewer Concerns:**

Here is a summary of the key concerns raised across different reviewers:
- Justifying the high-rank expressivity claim and MoSA's theoretical mechanism
- Providing formal derivation and clarifying the constrained optimization perspective
- Evaluating on more diverse model backbones (e.g., Qwen family)
- Including stronger and more recent PEFT baselines (e.g., LoRA-GA, PiSSA)
- Quantifying training latency and memory overhead compared to other methods
- Investigating sensitivity to key hyperparameters like the learning rate

The authors have provided substantial additional experiments and explanations, which adequately address most of the reviewers' concerns.

**Reviewer Scores:**

This paper received four reviews, initially split between two borderline accept and two borderline reject recommendations. During the discussion phase, one of the dissenting reviewers changed their assessment to borderline accept, resulting in a final consensus of three borderline accepts. The other dissenting reviewer did not participate in the discussion.

The authors have satisfactorily addressed the reviewers' primary concerns in their response. Therefore, the Area Chair recommends acceptance.

---

### Decision · Program_Chairs · 2026-01-26

Accept (Poster)